# Reviews and syntheses: Influences of landscape structure and land uses on local to regional climate and air quality

Raia Silvia Massad[1], Juliette Lathière[2], Susanna Strada[3], Mathieu Perrin[4], Erwan Personne[1], Marc Stefanon[5], Patrick Stella[4], Sophie Szopa[2], Nathalie de Noblet-Ducoudré [2]

1 UMR ECOSYS, INRA AgroParisTech, Université Paris Saclay, 78850, Thiverval Grignon, France
2 Laboratoire des Sciences du Climat et de l'Environnement, LSCE/IPSL, CEA-CNRS-UVSQ, Université Paris-Saclay, Gif-sur-Yvette, 91191, France
3 The Abdus Salam International Centre for Theoretical Physics - Earth System Physics Section, 34151 Trieste, Italy
4 UMR SAD-APT, AgroParisTech, INRA, Université Paris-Saclay, 75005, Paris, France
5 LMD/IPSL, Ecole polytechnique, Université Paris-Saclay, Sorbonne, Universités UPMC Univ. Palaiseau France

*Correspondence to*: Raia Silvia Massad (raia-silvia.massad@inra.fr)

**Abstract**. The atmosphere and the land surface interact in multiple ways, for instance through the radiative-energy balance, the water cycle or the emission-deposition of natural and anthropogenic compounds. By modifying the land surface, land-use and land-cover changes (LULCCs) and land management changes (LMCs) alter the physical, chemical and biological processes of the biosphere and therefore all land-atmosphere interactions, from local to global scales. Through socio-economic drivers and regulatory policies adopted at different levels (local, regional, national or supranational), human activities strongly interfere in the land-atmosphere interactions, and those activities lead to a patchwork of natural, semi-natural, agricultural, urban and semi-urban areas. In this context, urban and peri-urban areas, which have a high population density, are of particular attention since land transformation can lead to important environmental impacts and affect the health and life of millions of people. The objectives of this review is to synthesize the existing experimental and modelling works that investigate physical, chemical and/or biogeochemical–interactions between land surfaces and the atmosphere, therefore potentially impacting local/region climate and air quality, mainly in urban or peri-urban landscapes at regional and local scales.

The conclusions we draw from our synthesis are the following. (1) The adequate temporal and spatial description of land-use and land-management practices (e.g. areas concerned, type of crops, whether or not they are irrigated, quantity of fertilizers used and actual seasonality of application) necessary for including the effects of LMC in global and even more in regional climate models is inexistent (or very poor). Not taking into account these characteristics may bias the regional projections used for impact studies. (2) Land-atmosphere interactions are often specific to the case study analysed; therefore, one can hardly propose general solutions or recommendations. (3) Adaptation strategies, proposed after the evaluation of climatic impacts on the targeted resource have been derived, but are often biased as they

do not account for feedbacks on local/regional climate. (4) There is a need for considering atmospheric chemistry, through land-atmosphere interactions, as a factor for land-management, helping to maintain air quality and supporting ecosystem functioning. (5)There is a lack of an integrated tool, which includes the many different processes of importance in an operational model, to test different land use or land management scenarios at the scale of a territory.

## 1.    Introduction

The Earth's atmosphere is an envelope of gases, bearing liquid and solid particles that provides essential conditions for life to thrive on Earth. Via its composition and exchanges with the land surface, the Earth's atmosphere regulates the physical climate around us, is as a non-dissociable part of every ecosystem and a limited resource. Nowadays, facing global changes in terms of climate, atmospheric composition, biodiversity and demography, there is a growing demand to preserve a standard quality of life. On the other hand, there is a raising pressure on natural and man-shaped ecosystems to increase production and meet the nutritive and recreational demands of an expanding population. To maintain liveable conditions on Earth, it is important to understand the delicate balance between physical, chemical and biological processes, and their interactions, that involve the atmospheric envelope and related surface systems (water, soil, flora, fauna, concrete …) at local, regional, and global scales.

The atmosphere and the land surface interact in multiple ways, such as through the radiative-energy balance (Suni et al., 2015), the water cycle (Pielke et al., 1998), or the emission-deposition of natural and anthropogenic compounds (Arneth et al., 2010). Land-use and land-cover changes (LULCCs) (e.g., deforestation/afforestation, urbanization, cultivation, drying of wetlands, etc.) and land management changes (LMCs) (e.g., no-till agriculture, double-cropping, irrigation, cover crops, etc.) alter the land surface by modifying the physical properties (e.g., surface albedo, emissivity, and roughness), the chemical emission/deposition potential of land surfaces, and the biological equilibrium of living organisms and soils. Finally, LULCCs and LMCs affect the physical and chemical interactions between the land surface and the atmosphere, the atmospheric composition, and lastly the Earth's climate (Perugini et al., 2017), at scales spanning from local to global ones. The importance of LULCCs on the global climate is widely acknowledged, and global climate models (GCMs), which work at scales of 50-100 km, now integrate LULCC scenarios to investigate future climates (Jones et al., 2014). However, there is a raising need to understand the effects on climate of LULCCs and LMCs operating at the regional, local and even territorial scales, and hence to implement LULCC and LMC scenarios in climate models working at finer resolutions (i.e., regional climate models, RCMs) to explore their effects on the regional-local climate.

Nowadays, human activities largely shape landscapes, resulting in a patchwork of natural, semi-natural, agricultural, urban and semi-urban/peri-urban areas at scales smaller than hecto kilometres (Allen et al., 2003). The land surface is thus strongly sensitive to socio-economic drivers and influenced by regulatory

policies adopted at the local, regional, national or supranational scales. In this way, human activities strongly interfere in the land-atmosphere interactions and consequently influence climate and air quality at various geographical scales.

Recently, several reviews have examined the interactions between LULCCs and air quality and/or climate change.

Pielke et al. (2011) and Mahmood et al. (2014) reviewed the direct influence of LULCCs on regional climate, through biophysical processes, i.e. the modification of the water, energy and radiative exchanges between the Earth's surface and the atmosphere's lower boundary from local to regional scales. Based on both observed and modelled data, the authors conclude that LULCCs affect local and regional climate, and, more significantly, the areal coverage of the landscape conversion determines the potential of LULCCs to effectively influence the mesoscale and regional climate.

Arneth et al. (2010 and 2012), and more recently Heald and Spraklen (2015), mainly focused on the chemical effects. Arneth et al. (2010) looked at the picture from a global perspective with no special focus on LULCCs. They put forward that feedbacks between the terrestrial biosphere and the atmosphere cannot be ignored from a climate perspective, and that our limited understanding of the processes involved implies that none of the feedbacks studied will act in isolation but rather that the system is more complex. The authors warned that non-linearities and possible thresholds exist that should be elucidated before performing simulations with ecosystem–chemistry-climate models. Arneth et al. (2012) that encourage to improve the representation of biological and ecological processes and to bridge the gap between biogeophysical and socio-economic communities corroborate the need for integrative investigations. Indeed, the authors claim that the level of description for the different processes and interactions involved can significantly modify the projections of land-atmosphere exchanges (physical and chemical) performed with models.

Heald and Spraklen (2015) reviewed the interactions between LULCCs and atmospheric chemistry, with a focus on short-lived atmospheric pollutants, mainly biogenic volatile organic compounds (BVOCs), soil nitrogen oxides ($NO_x$), dust, smoke, bioaerosols, and ozone ($O_3$), and their subsequent radiative effects on global and local climates. The authors estimates that LUC can cause a regional direct radiative effect of $\pm20$ W m$^{-2}$. They identified several gaps of knowledge particularly linked to the aerosol effects on the regional radiative balance and emission variability due to different vegetation types. Other identified uncertainties are the future evolution of agricultural practices as well as the lack of connection between the different atmospheric species, or process responses to, LULCCs.

More recently, some studies have focused on the impact of small-scale changes, especially urbanization, on climate and air quality. The work led by Jacobson et al. (2019), for instance, investigated the impact of urbanization in two cities, New Delhi and Los Angeles, on weather, climate and air quality over the 2000-2009 period. The authors applied satellite and road data to assess the extension of urban and road areas, 1-year inventory for anthropogenic and natural emissions, together with a global-through-urban nested climate-weather-air pollution model (GATOR-GCMOM). Changes in natural emissions related

to meteorology were accounted for in this approach. For both New Delhi and Los Angeles, they concluded that urbanization has led to an increase in surface roughness, shearing stress and vertical turbulent kinetic energy, and concurrently to a decrease in near-surface and boundary layer wind speed, thus worsening pollution levels. This study shows that urbanization could have had significant impacts on both meteorology and air quality. Putting these results in a larger regional context would give the possibility to quantify the impact of urbanization on air quality and climate of surrounding peri-urban and rural areas. In that respect, Zhong et al. (2018) investigated the impact of urbanization-induced land-cover change and increase in anthropogenic emissions on the air quality of the megacity cluster of the Yangtze River Delta. The authors applied a regional climate-chemistry model (the Weather Research and Forecasting with Chemistry, WRF-Chem) coupled with an urban canopy model. A strong reduction of near-surface aerosol concentrations was estimated over urban regions, whereas particulate pollution increase over the surrounding rural areas. These results were partly due to the urban heat island effect, which increased the lower atmospheric instability and ventilation over the urban area, and therefore promoted the dispersion of pollutants from urbanized areas to their immediate vicinities. This study exhibits the tight links between processes (physical, chemical) and scales (local, regional; urban, peri-urban and rural areas).

So far, beyond scientific literature, relatively little attention has been paid in spatial planning practices to the consequences of land-use related decisions and measures on climate conditions and air quality at a local-regional scale. Spatial-planning concerns generally focus on the impacts of densely built-up areas on temperatures in urban contexts (Tam et al., 2015; Du et al., 2007), or on ways to improve the mitigation of climate change (i.e., to enhance the biospheric sink of carbon dioxide, $CO_2$, or decrease its sources). Hence, to our knowledge, very few studies have (1) discussed altogether the different physical, chemical and biological interactions between the land surface and the atmosphere, (2) focused on urban/peri-urban areas at local-regional scales, and (3) been addressed to decision makers, stakeholders and land planners.

Our objective is therefore to review the existing experimental and modelling works that investigate the effects of regional and/or local LULCCS and LMCs on physical, chemical and/or biological interactions and feedbacks between the land surface and the atmosphere in rural, urban and/or peri-urban landscapes. We refer to biological interactions as the exchange of chemical compounds that involve soils and biological organisms. The structure and content of this review is designed to be accessible to a large audience, including both specialists, such as scientists, and non-specialists, such as land-planners, stakeholders and decision makers. Non-specialists may refer to the appendix for a short review of the fundamentals of physics, chemistry and biology that are at work in LULCCs and LMCs.

Our synthesis focuses on relatively short time scales (with respect to climate), ranging from a few days to a few years, and on local to regional spatial scales going from a few to a hundred kilometres. In the text, we will abusively use the word 'climate' to refer to changes in mean weather values, considering impacts on local and meso-climate, whereas LULCC-induced impacts on global climate, especially via

modification in greenhouse gas emissions and concentrations, are not the target of the present study and will not be covered here. Readers interested in these topics may refer to the studies of Le Quéré et al. (2018) and Saunois et al. (2018) for example. We put a special attention to the territorial dimension, understanding territory as the portion of the land surface delimited and developed by a community according to their needs; this includes the political authority as well as the use and developments made by a social group (Le Berre, 1992, Ginet, 2012). We mainly focus on human-driven changes to land use and land management and on peri-urban landscapes, relying on the fact that today 54% of the world's population lives in cities (United Nation, 2014) and that the annual rates of urban land expansion ranges from 2.2% in North America to 13.3% in coastal areas in China. Although nowadays urban areas represent less than 0.5% of the Earth's total land area (around 650 000 km²) (Schneider et al., 2009), estimations show that more than 5.87 million km² of land are likely to be converted into urban areas by 2030, and very likely (probability >75%) for 20% of this surface (Seto et al., 2012).

We firstly present land-atmosphere interactions for individual land cover and/or land management changes by discussing physical, chemical and biological processes. We then explore possible interactions between these processes for a mosaic of different adjacent land uses and managements. We finally identify challenges and needs for current research and propose potential levers for action.

## 2. Land Cover and Land Use changes: history, dynamics and challenges

### 2.1. Historical perspective

Historically, research on land-use intensification and population growth emerged after World War II in different disciplines such as human geography, ecological anthropology or political ecology and concentrated on understanding agricultural changes. Later, concerns have been raised about the influence of the land surface on climate processes. In the mid-1970's, diverse studies highlighted the impact of land-cover change on the land-atmosphere energy balance at local, regional and global scales due to modifications in surface albedo (Ottermann, 1974; Charney and Stone, 1975; Charney, 1975; Charney et al., 1977; Sagan et al., 1979). Lately, in the early 1980's, Woodwell et al. (1983) and Houghton et al. (1985, 1987) emphasized the role of terrestrial ecosystems as sources and sinks in the carbon cycle, pointing out the impact of the land-cover on global climate. Because of the growing awareness that land surface influences various environmental processes and the climate, understanding the trends, patterns and mechanisms of LULCCs became a fundamental issue in academic research (e.g., Ramankutty and Foley, 1999; Klein Goldewijk, 2001; Foley et al., 2005; Lambin et al., 2006; Klein Goldewijk et al., 2011, Ellis, 2011). In the 1990s, the availability of land-use data through remote sensing shifted the focus from land-use intensification to land use and land cover studies (Erb et al., 2007; Verburg et al., 2011). More recently, due to its far-reaching, potentially detrimental ecological consequences, land-use intensification has attracted the interest of the scientific community at large (Erb et al. 2013).

## 2.2. Land Use and Land Cover Change

Although land change may be one of the most ancient of all human-induced impact on the environment, the Earth's land surface has never been altered by anthropogenic activity at the pace, magnitude and extent experienced over the past few centuries (Lambin et al., 2001). On the basis of distinct studies, it can be estimated that roughly 12 million km² of forests and woodlands have been cleared over the last three centuries, representing approximately a 20% decrease in the forest cover: Richards (1990) estimated a 11.7 million km² loss - from 62.2 to 50.5 million km² - between 1700 and 1980, Ramankutty and Foley (1999) indicated an 11.35 million km² loss - from 55.27 to 43.92 million km² - between 1700 and 1992, while Klein Goldewijk (2001) mentioned a 12.9 million km² loss - from 54.4 to 41.5 million km² - between 1700 and 1990. Although huge variations can be noticed between studies, notably because of land-use/cover definition and classification issues, similar trends have been reported regarding changes in natural areas (steppes, savannas, grasslands, shrublands, tundras and hot/ice deserts): Ramankutty and Foley (1999) mentioned a 7.3 million km² loss - from 73.2 to 65 million km² - between 1700 and 1992, while Klein Goldewijk (2001) assessed a 25 million km² loss - from 71.9 to 46.9 million km² - between 1700 and 1990. In his review on the anthropogenic transformations of the terrestrial biosphere, Ellis (2011) spatially quantified the temporal aspects of human transformations on the ecosystems (Figure 1).

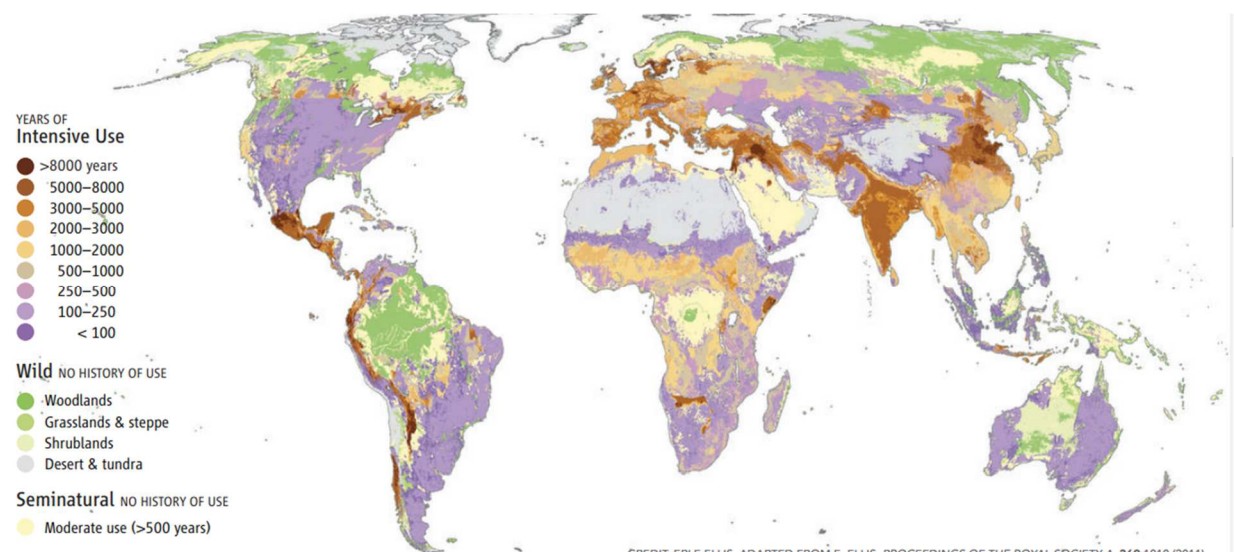

**Figure 1: Anthropogenic transformation of the terrestrial biosphere showing the number of years of intensive use from Ellis (2011).**

Such a focus has led to consider, especially under the scope of an integrated land science, the various and complex interactions between human societies and the environment (Turner, 2002). The land-cover - which can be understood as one biophysical attribute of the surface (Turner et al., 1995) - is now predominantly dependent on the land-use - which can be understood as the activity human societies have decided on the land in accordance with economic, cultural, political, historical and land-tenure considerations (Turner et al., 1995). On a world's ice-free land surface of approximately 130.1 million

km², the area directly reconfigured by human action as of 2007 has been estimated at 53.5 % (Hooke and Martin-Duque, 2012).

This decline of natural ecosystems is essentially due to the conversion of forests, savannas, and grasslands into agricultural lands. The global areas of croplands and pastures increased significantly since 1700 with estimated extension from 12.3 million (Goldewijk et al., 2011, 1700-2000 period) to 14.75 million km² (Pongratz et al. 2008; 1700-1992 period). By combining the results of different studies addressing this land transformation issue, Hooke and Martin-Duque (2012) estimated that, nowadays, croplands and pastures represent, respectively, 12.8% and 25.8% of the world's ice-free land surface (Figure 2).

Finally, the land transformation related to urban development and infrastructure expansion must be pointed out. A total of 8.4 million km² can be classified as urban areas, rural housing, business areas, highways or roads (Hooke and Martin-Duque, 2012). Even if increasing, commonly at the expense of agricultural land (Döös, 2002), this number represents only about 6.46% of the world's ice-free land surface. However, such a land transformation can strongly affect environmental processes at local and/or regional scales and therefore affect the health and life of million people, given the human density in the areas impacted (Ermert et al., 2012; Jagger and Shively, 2014).

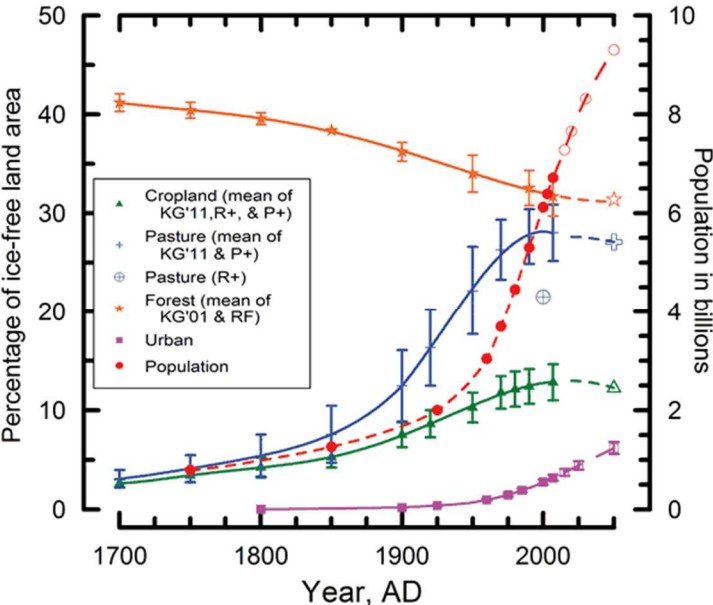

**Figure 2: Adapted from Hooke and Martin-Duque (2012) Changes in land use through time (closed symbols) with extrapolations to 2050 AD (open symbols).**

**2.3. Land-use Intensification**

Another aspect of land-use that affects the environment is land-use intensification. In the scientific literature, there is no unique definition of land-use intensification or land-use intensity, even though the concept is increasingly referred to. The diversity of definitions reflects on one hand a disciplinary diversity and, on the other, a certain relationship between man and nature (Lindenmayer et al., 2012, Erb et al., 2013, Erb et al., 2016). From these two different contexts two distinct definitions of land use

intensification emerge. The first comes from an agricultural point of view where land use intensification is simply defined as the increasing production from the same land by additional inputs in terms of labour, energy, fertiliser and water (Erb et al., 2009; Krebs et al., 1999). Most of the time this involves developed agricultural techniques and an increased amount of inputs to the ecosystem (fertilisers, pesticides, etc.)

(Lindenmayer et al. 2012). The land-use intensification via production is thus operated in a neutral way on land area where intensification is the means by which gains are made using increased inputs per unit land area (Moller et al., 2008). However, it can involve a land use change in the case of the implantation of bio-energy crops, for example. As a second definition, land use intensification can also be seen from an ecological or biodiversity point of view as the increasing transformation of the land away from the

original habitat. From this point of view, land use intensification is accompanied by landscape and ecosystem simplification, from complex natural systems to simplified agricultural ecosystems (the more one goes in intensification, the more the other tends to go towards landscape uniformity in a reduction of biodiversity), or to urbanisation (Flynn et al., 2009). This type of intensification is, however, never neutral on land area and systematically involves a LULCC. The difference between this view on land-

use intensification and LULCC is that the change is always towards a more man-shaped system, whereas LULCC can occur in the opposite direction by afforestation, restoration etc.

As a result, it is very difficult today to draw a picture of the dynamics behind or the evolution of land use intensification simply because there is no common definition and terminology and there are many knowledge gaps related to the underlying processes and determinants of the levels, patterns and

dynamics of land-use intensity (Shriar, 2000; Erb, 2012). However, it is essential to a) assess the impacts of those changes and intensifications and b) have the tools to assess their influences on the biosphere as on biosphere-atmosphere interactions. In the sections below, we review the documented effects on the atmospheric compartment from a physical, chemical and biological point of view and classify them in two categories (Figure 3): land cover change and land intensification (agricultural and urban).

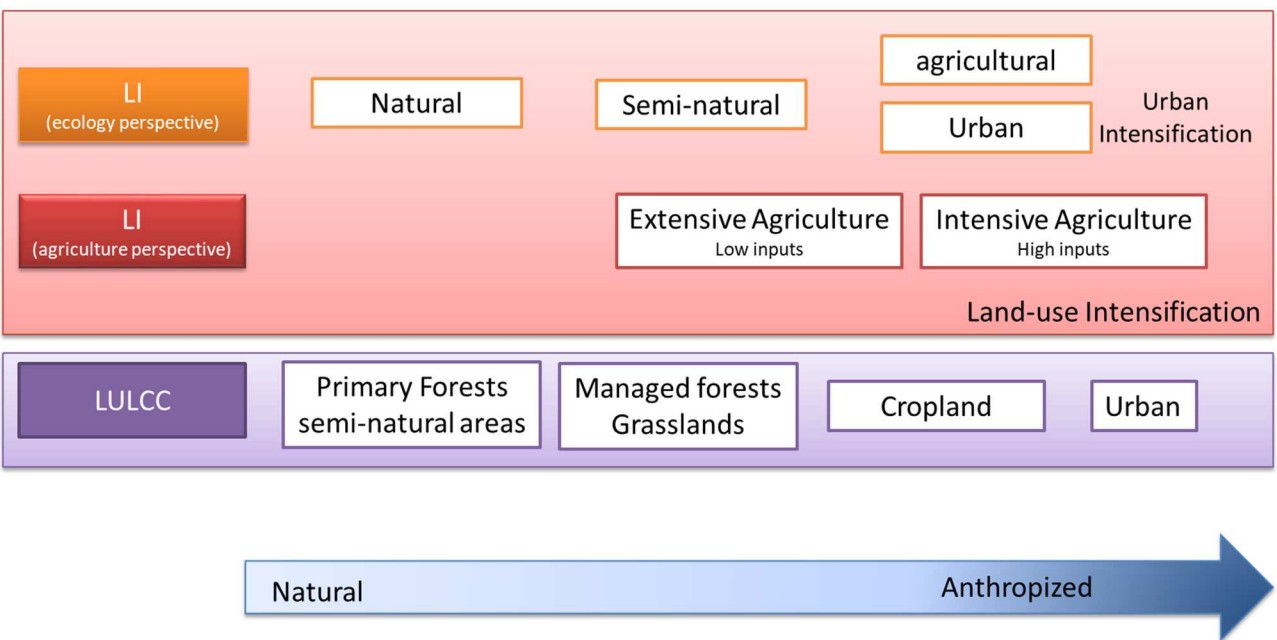

**Figure 3: Main changes in LULCC and LI (land-use intensification) from an anthropic perspective and their classification relative to the sections of this manuscript**

### 3. Human driven land use and land management changes and their impact on climate and air quality

#### 3.1. Land Cover Change

Most historical LULCCs are considered to have globally decreased primary production and therefore had an impact on atmospheric $CO_2$ concentrations and thus on global warming, as shown by Gruber and Galloway (2008). This can be explained by the fact that past LULCCs concerned primarily deforestation and the increase of urban areas, thus leading to lower ecosystem productivity and a release of soil and biomass stored carbon to the atmosphere in the form of $CO_2$. Moreover, LULCCs affects physical interactions between the land-surface and the atmosphere and atmospheric components other than $CO_2$ such as reactive nitrogen compounds via their effects on the carbon (C) and nitrogen (N) cycles. This is mainly induced by the alteration of land-atmosphere exchanges through changes in (i) stomatal conductance, (ii) deposition and adsorption on the leaf surfaces and cuticles, which varies according to plant species, (iii) the canopy architecture and its physical properties (leaf area, tree height), and (iv) availability of free soil water, which affects the production and the exchange of certain compounds, as illustrated below by some examples.

##### 3.1.1. Deforestation/Afforestation

Deforestation had been practiced for tens of thousands of years for agriculture, grazing, cultivation and urban purpose. However, over the last three centuries deforestation drastically has increased, with around 12 million km² of forests cleared and 40 million km² remaining today (Ramankutty and Foley, 1999; Klein Goldewijk, 2001; http://www.fao.org/forestry/fra/41256/en/).

From a **Physical** perspective: Several studies investigated the effects on climate of deforestation, or of its opposite (afforestation) mainly via a modelling approach. These studies compare the effects on climate of changes between current and pre-industrial potential vegetation, under the hypothesis of no human activities. Among its biogeophysical effects on climate, deforestation has contrasting effects on air temperature that depend on the latitude and the vegetation types involved (Claussen et al., 2001; Snyder et al., 2004, Gibbard et al., 2005; Bala et al., 2007; Betts et al., 2007; Jackson et al., 2008; Davin and de Noblet-Ducoudré, 2010; Beltràn-Przekurat et al., 2012). At high latitudes, deforestation triggers a winter and spring surface cooling due to changes in the radiation budget that compensate, at the annual scale, the summer warming resulting from decreased latent heat flux (i.e. evaporation). In particular in boreal regions, forest removal strongly increases the surface albedo. Indeed forests mask the snow as opposed to herbaceous vegetation (Chalita and LeTreut, 1994; Betts et al., 2001; Meissner et al., 2003; Randerson et al., 2006). At low latitudes, deforestation leads to a surface warming due to changes in the water cycle. Conversion of tropical rainforests to pasture lands (as in the Amazonia Basin region) strongly modifies surface evapotranspiration and roughness since, compared to pasture lands, trees have a higher surface roughness that enhances surface fluxes and thus the evapotranspiration cooling efficiency (Shukla et al., 1990; Dickinson and Kennedy, 1992; Lean and Rowntree, 1997, von Randow et al., 2004, Nogherotto et al., 2013; Lejeune et al., 2015; Spracklen and Garcia-Carreras, 2015; Llopart et al., 2018). In the long term, reduced evapotranspiration and precipitation may lengthen the dry season in the tropics, thereby increasing the risks of fire occurrence (Crutzen and Andreae, 1990). At mid-latitudes, both albedo and evapotranspiration mechanisms are at work and compete against each other, as recently confirmed by satellite-based observation analysis (Li et al. 2015, Forzieri et al. 2017). Although studies over the mid-latitudes show somewhat contradictory results and the effect on air temperature (warming/cooling) remains unclear in temperate regions such as the Mediterranean Basin region and Europe (Gaertner et al., 2001; Heck et al., 2001; Anav et al., 2010; Zampieri and Lionello, 2011; Gálos et al., 2013; Stéfanon et al., 2014; Strandberg and Kjellström, 2019), in the Northern hemisphere, the historical land-cover change has very likely led to a substantial cooling (Brovkin et al., 1999, 2006; Bonan, 1997; Betts, 2001; Govindasamy et al., 2001; Bounoua et al., 2002; Feddema et al., 2005a), comparable in magnitude with the impact of increased greenhouse gases (Boisier et al., 2012; de Noblet-Ducoudré et al., 2012). However, a recent study combines present-day observations and state-of-the-art climate simulations and show that historical deforestation in North America and Eurasia made the hottest day of the year warmer since pre-industrial time, contributing to at least one-third of the local present-day warming of the heat extremes (Lejeune et al., 2018). In addition to modify mean and extreme temperatures, deforestation/afforestation can also modify the hydrological cycle by enhancing or inhibiting convective clouds and precipitation in the overlying atmospheric column. Some studies show an enhancement of shallow cumulus clouds over deforested lands in Amazonia (Chagnon et al., 2004, Wang et al., 2009), while opposite results were

found over deforested lands in Southwest Australia (Ray et al., 2003). Two different mechanisms result from the interplay between the surface heat fluxes and the boundary layer structure (i.e., stability, temperature and humidity): (1) Dry soil and high sensible heat flux can increase the entrainment of cold air from the boundary layer top and finally increase shallow cloud cover by lowering the saturation threshold (Westra et al., 2012; Gentine et al., 2013); (2) On the contrary, wet soil and high latent heat flux moisten the boundary layer and increase the relative humidity at its top in case of deforestation.

From a **biological** perspective: Deforestation implies modifications in surface moisture and temperature that in turn might affect directly or indirectly decomposition rates and nutrient mineralization in soils (Dominski, 1971; Stone, 1973; Stone et al., 1979; Classen et al., 2015; Manzoni et al., 2012; Chen et al., 2014; Townsend et al., 2011; Bonan, 2008). As a result, both carbon and nitrogen release to the environment are forecasted to increase. The forest floor decomposes rapidly (Covington, 1976; Bormann and Likens, 1979) and, without forest regeneration, will eventually be partially eroded. The combination of increased decomposition (which consumes oxygen) and wetter soils (which slow oxygen diffusion) may also increase the occurrence of anaerobic microsites within the soils, which might contribute to methane ($CH_4$) emissions (Adji et al., 2014; Jauhiainen et al., 2016). Nitrogen can be lost to the atmosphere through ammonia ($NH_3$) volatilization, nitrous oxide ($N_2O$) production during nitrification (Bremner and Blackmer, 1978; Veldkamp et al., 2008), or denitrification to $N_2O$ or atmospheric nitrogen ($N_2$) (Firestone et al., 1980; Neill et al., 2005; Lammel et al., 2015). Soil properties such as soil organic carbon or soil nitrogen cycling respond to deforestation with a large spatial variety from one system to another (Powers and Schlesinger, 2002; Chaplot et al., 2010; de Blécourt 2013). However, the largest emissions of non-$CO_2$ greenhouse gases will probably result from agricultural use and management on deforested areas.

Finally, several studies show that there are feedbacks between tropical forests and climate change (Bonan, 2008). Carbon dioxide fertilization, for example, could have a positive effect by sustaining tropical forest growth (Lapola et al., 2009; Salazar and Nobre, 2010). This is exacerbated by N fertilization effect since tropical areas are no-limited N environments and N is increasing through atmospheric deposition in non-tropical areas (Magnani et al., 2007; Sutton et al., 2008; Samuelson et al., 2008; Jackson et al., 2009). Zaehle et al. (2011) showed that N inputs increased C sequestration by ecosystems, and Churkina et al. (2007) attributed 0.75–2.21 GtC yr$^{-1}$ during the 1990s to regrowing forests. However Yang et al. (2010) showed that the contribution of N fertilization is lower for secondary forests regrowth (Jain et al., 2013).

As a direct effect, afforestation inevitably leads to carbon loss from the system (Feddema et al., 2005b; Foley et al., 2005; Le Quéré et al., 2012; Houghton et al., 2012). However, large uncertainties remain on i) how these altered ecosystems will react to induced global climate change (increased

$CO_2$ concentration, increased temperature, etc.), ii) changes in the emissions of non-$CO_2$ greenhouse gases ($N_2O$, $CH_4$) and iii) changes in the exchange of reactive trace gases.

From a **chemical** perspective: afforestation directly affects BVOC emissions, since trees are high BVOC emitters, as documented by Purves et al. (2004) over the Eastern U.S. by combining a BVOC emission model with vegetation changes as recorded by the USDA Forest Service Inventory Analysis (FIA) over surveyed forest plots. Over the target region, emissions of the main BVOCs (i.e., isoprene and monoterpene) have increased, especially under heatwave conditions (i.e., daily air temperature above 35° C), due to increase in the forest leaf area mainly driven by human disturbance via harvesting and plantation management (i.e., often plantation forestry introduces high-emitters), but as well by perturbing ecological succession with fires and pollution. Enhanced BVOC emissions from forests are likely to modify the $NO_x$-VOC-$O_3$ regime, nevertheless the outcome critically depends on the fate of isoprene nitrates, whether they are a terminal or temporal sink of $NO_x$ (Val Martin et al., 2015). Concerning fine-mode aerosols, summer levels of $PM_{2.5}$ (i.e., particulate matter, PM, with aerodynamic diameters $\leq 2.5$ μm) are predicted to increase with afforestation due to the formation of BSOAs from BVOCs (Heald et al., 2008; Trail et al., 2015; Val Martin et al., 2015). As afforestation, deforestation to create pasture or crop lands can as well exacerbate $O_3$ levels by increasing $NO_x$ emissions from soil microbial activity, promoted with fertilization (Ganzeveld and Lelieveld, 2004; Trail et al., 2015); in winter, the enhanced NOx levels favour nitrate aerosol production, while in summer deforestation decreases aerosol deposition, by reducing surface roughness. In conclusions, fine-mode aerosols such as $PM_{2.5}$ may increase all year-round under deforestation (Trail et al., 2015).

Under a raising demand and interest for fast-growing plants for food production, cattle feed, domestic products and biofuels, plantation are rapidly expanding all over the world. The choice of crop or tree type influences BVOC emissions and the resulting $O_3$ and BSOA levels (Hewitt et al., 2009; Ashworth et al., 2012; Warwick et al., 2013; Stavrakou et al., 2014). This is the case of oil palm crops that show much larger BVOC emission potentials compared to primitive forests (from 3 to 10 times higher for isoprene, Hewitt et al., 2009 and Fowler et al., 2011). In South-East Asia, increasing BVOC emissions from oil palm plantation interplay with raising NOx emissions resulting from the spread in mechanization, fossil fuel use, and fertilizer application associated with the oil palm industry. The complex interaction between BVOC and NOx finally enhances O3 levels at local-regional scales (Goldammer et al., 2009; Hewitt et al., 2009; Silva et al., 2016; Harper and Unger, 2018), with even trans-boundary effects (i.e., downwind regions) (Warwick et al., 2013). Similarly, to South-East Asia and oil palm production, the expansion of biofuel production in Europe could modify future LULC to satisfy the increasing demand for renewable energy sources (Beringer et al., 2011). Among biofuel feedstock, crops as miscanthus or 2nd generation plantation such as poplar show higher isoprene emission potential compared to European native species. The conversion of European grass- and crop-lands into biofuel plantations may affect summer $O_3$ levels

with effects that strongly depend on the interaction between BVOC and $NO_x$ emissions. For example, to limit the effects on $O_3$ production of a steep increase in isoprene emissions (~45%) from conversion of 5% of European grass- and crop-lands into poplar plantation, $NO_x$ emissions should be reduced by 15-20% (Beltman et al., 2013). Regarding Europe, Ashworth et al. (2013) showed that the extension of short-rotation coppice for biofuel feedstock could have small but yet important impacts on surface $O_3$ concentrations, and subsequently on human mortality and crop productivity, since it would modify emitted compounds and their levels. Being BSOA precursors, enhanced BVOC emissions from afforestation are also involved in particulate matter pollution.

Using a large-scale chemistry-transport model for present-day climate, Ashworth et al. (2012) investigated the impact of realistic large-scale scenarios of biofuel feedstock production (~100 Mha plantations) in both the tropics and the mid-latitudes on isoprene emissions, $O_3$ and BSOA formation. These LULCCs drive an increase in global isoprene emissions of about 1%, with substantial impact on regional $O_3$ levels and BSOAs. In the tropics, the expansion of oil palm plantations enhances BSOAs by 0.3 μg m$^{-3}$ (+3-5%, BSOA annual mean concentrations: 6–10 μg m$^{-3}$). In the mid-latitudes, the establishment of short-rotation coppice increases BSOA concentrations up to 0.5 μg m$^{-3}$ (+6%, from 8 μg m$^{-3}$).

### 3.1.2. Wetland conversion / Restoration

Although wetland drainage is a relatively small proportion of the world's land surface, LULCC can have significant impacts on some areas. Wetland drainage for agriculture purposes has removed between 64–71% of natural wetlands since 1900 (Davidson, 2014).

From a **Physical** perspective: Only few studies evaluated its impact on local/regional climate. The most documented case is that of South Florida (Pielke et al., 1999; Weaver and Avissar, 2001; Marshall et al., 2004a, 2004b). During the 20[th] century, large wetland areas in South Florida were converted to large-scale crops (cereals), citrus growth, and fruit crops in general. Modelling studies show that current surface cover caused significant changes in temperature extremes with increased length of freezing events and increased magnitude of those frost (lower temperature), which severely reduced the agricultural production (Marshall et al., 2004a). During night time, water vapour evaporates from the swamps and modifies the longwave radiation budget, resulting in a less rapid infrared cooling and less cooling by +2°C than for the current (drained) case. A similar study over Switzerland shows opposite results (Schneider and Eugster, 2007). The conversion of wetlands to extensive farming caused a night time warming and a daytime cooling of a few tenths of a degree Celsius. This temperature modification was explained by the alteration of soil thermal properties and by higher albedo in the current case. During the night time, higher thermal conductivity of the current soils resulted in upward heat fluxes, which enhanced the temperature. In another vein, Mohamed et al. (2005) studied the effect of Sudd swamp on the Nile water flow and local climate. Due to the Sudd wetland, located in the upper Nile, a substantial amount of water is lost through

evapotranspiration. In a drained Sudd scenario produced by a numerical experiment, the Nile flow just downstream the wetlands increases by 46 $Gm^3yr^{-1}$ over a total of 110 $Gm^3yr^{-1}$. However, evapotranspiration reduces, causing a temperature increase by +4-6°C during the dry season.

From a **Biological** perspective: The drainage of peatlands and wetlands for agricultural use alters several characteristics of those areas and could thus be problematic (see Verhoeven et al., 2010 for a review). Especially in tropical areas, peatland draining releases some extra $CO_2$ by oxidizing and subsiding peat soils used for growing oil palms (Immirzi et al., 1992; Maltby and Immirzi, 1993; Safford et al., 1998; Furukawa et al., 2005). Hoojer et al. (2006) estimate to 516 Mt C $yr^{-1}$ the emissions from Indonesian peatland draining (fires excluded). On the other hand, since wetlands are a considerable source of $CH_4$, their drainage will decrease emissions of $CH_4$ and can thus be considered a carbon gain from that point of view (Bergkamp and Orlando, 1999; Maltby and Immirzi, 1993). However, this gain is counterbalanced by increased $N_2O$ emissions, due to the lowering of the water table (Kasimir-Klemedtsson et al., 1997; Maljanen et al., 2010). On the other hand, changes in vegetation and therefore growth in those drained areas involve an increased carbon sink from vegetation. However, this additional sink rarely compensates for the GHG losses resulting from C losses from the soil (Yeh et al. 2010; Yew et al. 2010).

From a **chemical** perspective: On top of decreasing $CH_4$ emissions, wetland drainage may probably increase $NO_x$ emissions, and modify emissions of other compounds such as BVOCs, due to vegetation change, which together could contribute to significant changes in the atmospheric chemical composition. Overall, the impact of wetland conversion on compound emissions other than $CH_4$ and on atmospheric chemistry has been poorly investigated.

## 3.2. Land intensification

### 3.2.1. Urbanization

Urbanization results in the replacement of (pseudo-)natural ecosystems vegetation by more or less dense and impervious built-up environments. Human activities concentrated in these areas are responsible for additional heat and gaseous releases in the atmosphere. Consequently, these LULCCs sharply modify the atmosphere, both in terms of climatic conditions and gas composition, which ultimately affect land-atmosphere exchanges and biogeochemical cycles.

From a **Physical** perspective: Urbanization results in a modification of surface radiative budget, energy balance, water balance and land-atmosphere mass and energy exchanges (see equations 1 to 5 in the appendix), leading ultimately to (local-) climate alteration in urban areas.

Firstly, urbanization affects each components of the radiative budget. On one hand, the net radiation is potentially reduced due to the decrease in the incoming shortwave radiation that is screened out by a reflecting smog layer. In the dry season, on clear skies, Jauregui and Luyando (1999) observed that the incoming solar radiation over Mexico City was 21.6% lower than its suburbs. This difference could raise up to 30% under weak winds. However, the intensity of the reduction in

incoming short wave radiation was closely related to the day of the week (i.e., human activities) and meteorology (e.g., temperature, humidity, solar radiation), which both influence photochemical smog formation. Similarly, Wang et al. (2015) measured lower incoming short wave radiation in Beijing compared to its surrounding, with values ranging between 3% and 20% depending on the season. Focusing on summer periods (June, July, Augustt), Li et al. (2018) recorded lower S↓ at urban stations compared to rural stations in the city of Berlin; the authors attributed this dimming effect to the thick aerosol layer observed over the city. Based on the analysis of global radiation measurements from the Global Energy Balance Archive (GEBA), Alpert et al. (2005) and Alpert and Kishcha (2008) showed a relationship between solar dimming, population density and atmospheric pollution such as aerosols, which absorb and scatter the incoming solar radiation. Overall, Alpert and Kishcha (2008) demonstrated that at the surface short wave radiation is 8% lower in urban compared to rural areas. Moreover, the net radiation is also potentially reduced by the enhanced outcoming longwave radiation due to a warmer urban environment (the so-called "Urban Heat Island effect", see below) since infrared radiations depend on surface temperature. On the other hand, urbanization also induces an increase in net radiation. Urbanization usually results in a decrease of surface albedo ($\alpha$) and surface emissivities ($\varepsilon_s$) (Table 1), finally reducing both outgoing short and longwave radiations. Although some building materials exhibit larger albedo and emissivity than (pseudo-)natural environments, most of them have lower ones, especially asphalt or other dark materials (e.g., Li et al., 2013; Alchapar et al., 2014; Rahdi et al., 2014). Yet, at the city scale, outgoing short- and longwave radiations is scattered and absorbed multiple times within urban canyons (i.e., light trapping effect), thus contributing to both outgoing short- and longwave radiation reduction. Overall, both effects tend to compensate each other and only few differences in $Q*$ have been observed between urban and rural environments in yearly average (Oke and Fuggle, 1972; Christen and Vogt, 2004). Nevertheless, depending on the seasons and time of the day, larger net radiation have been observed in urban areas during daytime and in winter, when snow covers surrounding rural areas (Christen and Vogt, 2004).

Secondly, compared to the surrounding areas, urban environment sharply modifies the way surface energy is dissipated (i.e., the energy partitioning between sensible and latent heat fluxes). In rural environments vegetation and pervious surfaces provide larger evapotranspiration rates (i.e., latent sensible heat flux), therefore lower sensible heat flux), whereas in urban areas energy is mainly dissipated through sensible heat flux. A non-natural term sensible heat flux, due to heat release by human activities (e.g., building heating or cooling) adds to a natural sensible heat flux, further increasing sensible heat flux in urban areas (Arnfield, 2003). As a result, Bowen ratio amplifies in urban areas (Table 1). Such a large dissipation of energy through sensible heat flux, which transfers heat from the surface to the air, leads to the so-called "Urban Heat Island" effect (UHI), the most well-known alteration of (local-) climate due to urbanization that corresponds to a warmer climate in urban environments compared to surrounding rural environments (around 2-3°C). UHI is defined

as a temperature difference between the city and its surrounding, this last depending on the local land use. Nevertheless, UHI intensity is sharply variable according to the time of day (e.g., Pearlmutter et al., 1999), the season (e.g., Eliasson, 1996; Zhou et al., 2014), the geographical location, spatial organization of the urban fabric (e.g., building size and density, human use, fraction of vegetation) (e.g., Emmanuel and Fernando, 2007; Hart and Sailor, 2009), and rural land use (e.g., forests, crops, bare soil) (Chen et al., 2006). Recently, Yao et al. (2019) combined satellite-based observations of land surface temperature (LST) and enhanced vegetation index and showed that rural greening has contributed by +0.09°C per decade (23 %) over the period 200-2017 to the increasing in daytime surface UHI intensity (i.e., urban LST minus rural LST). By modifying the local energy budget, urbanization modifies the boundary layer structure and lastly influences the water budget. Urban signatures (e.g., change in magnitude, intensity and spatial patterns) have been observed in precipitation (see Shepherd et al., 2005 and Pielke et al., 2007 for a review on urban precipitation). Moreover, complex urban terrain amplifies regional gradients in temperature, pressure, moisture and wind that act as a source of vorticity for storm ingestion and development into tornadoes (Kellner and Niyogi, 2014). Moreover, urban areas can attenuate, split or deflect extreme storm events (e.g., Lorenz et al.,  2018), and modify their intensity and occurrence. Over the the Beijing metropolitan area, 60–95% of the selected weather stations show that the intensity and occurrence of extreme rainfalls slightly have reduced throughout 1975-2015, periods with consecutive rainy days (CRD) have lengthened, and the Julian dates of daily maximum precipitation have been delayed (Zhang et al., 2018). Furthermore, cities are important source of aerosols that help initiating thunderstorms (Haberlie et al. 2015). However, the joint study of UHI and urban pollution island is still in its infancy and the indirect radiative effect of aerosols (i.e., impact on cloud properties and formation) on UHI  need further investigations (Li et al., 2018).

To mitigate UHI-induced warming, vegetated or highly reflective roofs are being integrated in the built environment and have received a growing interest in climate modelling studies. Cool roofs absorb less incoming shortwave radiation than dark roofs. They decrease the local and regional summer surface temperature by 0.1-0.9°C (Millstein and Menon 2011 ; Georgescu et al., 2012 ; Salamanca et al., 2016 ; Vahmani et al., 2016). Their impact on climate is not just limited to surface energy budget as for example precipitation decrease was put forward in a modelling framework (Georgescu et al., 2012). Benefits from green roofs are analogous to cool roofs, as vegetation contributes to cooling via increased albedo, and water evapotranspiration. In situ experiments with different species have surface temperature difference up to 3°C (MacIvor and LundHolm 2011). However at the regional scale and over urban areas, simulated cooling is greater for the cool roofs relative to the green roofs, because of the vegetation seasonality and sensitivity to dryness (Georgescu et al., 2014).

From a **Biological** perspective: At a local scale, the development of urban areas and the related activities directly affect air quality and local temperatures, which leads to modifications in the

biology of organisms. Studies based on the analysis of tree traits along an urban – rural gradient showed that tree growth and phenology are affected by the vicinity of an urban area mainly due to increase in temperature (Gillner et al., 2014; Mimet et al., 2009; Dale et al., 2014), $CO_2$ concentrations (Calfapietra et al., 2010; Ziska et al., 2004), ozone deposition (Gregg et al., 2003; MacKenzie et al., 1995) and through the enhanced effect on air quality via the increased emissions of BVOCs (Calfapietra et al., 2013, Lathiere et al. 2006). Recent studies have also focused on the effects of soil waterproofing in urban areas that reduces water availability and exacerbates water stress in urban forests significantly affecting growth (Vico et al., 2014; Volo et al., 2014, Scalenghe and Marsan, 2009).

From a **Chemical** perspective: At local scales, urbanization directly affects both $O_3$ and aerosol levels by increasing the number of emission sources on a limited area (e.g., traffic, domestic heating). In the literature, there is a raising interest in the direct impacts of urbanization on air quality (special issues in the Atmospheric Chemistry and Physics journal related to the Megapoli-Paris 2009/2010 campaign, the MILAGRO and the CITYZEN projects, 2011; Baklanov et al., 2018, Zhu et al., 2019, Ooi et al., 2019), with a special focus on $O_3$ levels, summer pollution (Nowak et al., 2000; Civerolo et al., 2007; Jiang et al., 2008) and on the role of urban trees in $O_3$ pollution via BVOC emission changes (Chameides et al., 1988; Cardelino and Chameides, 1990; Corchnoy et al., 1992; Benjamin et al., 1996; Taha, 1996; Benjamin and Winer, 1998; Yang et al., 2005; Taha et al., 2015; Livesley et al., 2016; Churkina et al., 2017; Bonn et al., 2018).

Increase in urban LU following population growth exacerbates $O_3$ pollution during summer, mainly due to changes in $NO_x$ emissions (Zhu et al., 2019). In the greater Houston area (Texas), under a projected increase in urban LU by 62%, together with changes in anthropogenic and biogenic emissions, the number of extreme $O_3$ days in August rose by up to 4-5 days, with LUCs contributing to 2-3 days' increase (Jiang et al., 2008). In the greater New York City region, future urban LU changes may enhance episode-average $O_3$ levels by about 1-5 ppb, and episode-maximum 8h ozone levels by more than 6 ppb (Civerolo et al., 2007). In metropolitan regions, changes in $O_3$ levels show a heterogeneous spatial pattern: they decrease in the urban core, likely due to high $NO_x$ levels ($O_3$ titration), while they generally increase downwind of precursor sources (Civerolo et al., 2007; Jiang et al., 2008). In urban environment, BVOC emissions from urban trees seem to have negligible effect on summer $O_3$ levels (< 1 ppb compared to increases of 1-7 ppb due to urban LUCs; Nowak et al., 2000 vs. Jang et al., 2008). However, the effect of urban green areas on BVOC emissions and $O_3$ pollution depends on tree species (Taha et al., 1996, Taha et al., 2015); for this reason, the choice of urban trees based on their BVOC potential may be addressed as a critical urban land management practice (Benjamin et al., 1996; Benjamin and Winer, 1998; Churkina et al., 2015; Calfapietra et al., 2015; Grote et al., 2016). For example, in Beijing, deciduous trees dominate (76%) and some of the main species are high BVOC emitters (e.g., *Sophora Japonica* L., *Populus tomentosa* L., and *Robinia pseudoacacia* L.), that may favour a worsening in $O_3$ pollution due to the rapid increase in

$NO_x$ emissions (Yang et al., 2005). In Los Angeles metropolitan area, Corchnoy et al. (1992) measured BVOC emission rates of 11 tree species to underpin the selection of potential shade trees, whose planting should reduce the urban heat-island effect. Accounting for California climate, the authors suggested best (e.g., Crape myrtle and Camphor tree) and poor (e.g., Liquidambar and Carrotwood tree) choices for urban trees, and underlined that large difference in BVOC emissions should be factored into decision-making about shade trees to plant. In California's South Coast Air Basin, medium- and high-emitting trees may lead to hazardous $O_3$ levels (> 50 ppbv) (Taha, 1996). In the same geographical area, the most effective scenario to reduce the peak ozone involves replacing 4.5 Mha of high BVOC emitters with low BVOC emitters, while to target all-hour ozone the best choice consists in planting 2.5 Mha of low BVOC emitters in urbanizing areas and switching 4.5 Mha from high to low emitting species (Taha et al., 2015). It is important to remind that, although BVOC concentrations are usually lower than AVOC concentrations in urban areas, BVOCs react faster than AVOCs and can thus have significant effects in urban areas, as shown by Chameides et al. (1988) in the Atlanta metropolitan region.

At the regional scale, Chen et al. (2009) demonstrated that LULCCs can offset the impact of temperature on biogenic emissions and concluded that LULC evolution should be factored in the study of future regional air quality. Other than land-use, land-cover and land-management changes (LULC&LMCs) here discussed, changes in climate conditions and anthropogenic pollutant emissions (e.g., due to "clean air" policies) influence directly and indirectly air quality and interact in a non-linear fashion with LULC&LMCs, for this reason the climate-emission-land system should be consider as a whole when studying changes in surface $O_3$ and aerosols.

### 3.2.2. Agriculture Intensification

The main aim of agricultural management is to increase productivity and has therefore an immediate effect on the agricultural ecosystem functioning (Tillman et al., 2002). Most of these agricultural practices will also have direct or indirect impacts on the environment other than the biosphere (e.g., atmosphere, water, soils, etc.) (Sutton et al., 2011). Agricultural intensification also enhances the export of organic matter from the affected ecosystems with consequences such as the reduction of carbon and nitrogen cycling and soil degradation and erosion (Mattson et al., 1997; Ruysschaert et al., 2004). Examples of agricultural intensification are the conversion of pasture or grasslands into agricultural land, or including rotations of agricultural and grasslands.

### Irrigation.

From a **Physical** perspective: Among land-management practices, irrigation is one of the most common all over the world, and it significantly modifies the surface water and energy budget. The amount of additional water put into the soils tends to increase the latent heat flux at the expense of sensible heat flux, leading to an irrigation cooling effect (ICE) of the ambient air. In California, for example, this

effect was observed during daytime over a long-term dataset and estimated to several degrees (-1.8°C to -3.2°C since the beginning of irrigation - Lobell and Bonfils, 2007; Bonfils and Lobell, 2007). However, there are two opposite indirect heating effects. First, the high-albedo desert is converted into a low-albedo vegetated plain (Christy et al., 2006) which results from a combination of crop planting and irrigation and can therefore be classified as a land cover change rather than an agricultural intensification. Second, the greenhouse warming is enhanced due to the increase in water vapour. The greenhouse effect -less important than the transpiration effect on temperature- dominates during the night-time. Several modelling studies assess both greenhouse and transpiration effects (Boucher et al., 2004; Sacks et al., 2009; Puma and Cook, 2010; Cook et al., 2011, 2015; Kueppers et al., 2012) and highlight that locally the ICE may have partly masked the 20th century climate warming due to increased greenhouse gases (Kueppers et al., 2007). Meteorological studies suggest that irrigation can also lead to an increase in summer cloud cover and precipitation, as observed over the Great Plains region in the United States, downwind of the major irrigation areas (Segal et al., 1998; Adegoke et al., 2003; DeAngelis et al., 2010). In China, paddy cultivation requires water to stay on the ground during the rice-growing season leading to a moistening of the land surface, an increase of the latent heat flux and a decrease in the near-surface temperature from May to July in the Sichuan Basin (Sugimoto et al., 2019). Thiery et al. (2017) demonstrated that irrigation influences temperature extremes and lead to a pronounced cooling during the hottest day of the year (- 0.78 K averaged over irrigated land). Besides, this impact of irrigation on temperature is not limited to agricultural environment as the same cooling effect has been also reproduced for urban irrigation in a water-scarce region (Los Angeles area), with the largest influence in low-intensity residential areas (average cooling of 1.64 °C) (Vahmani and Hogue, 2015). Affecting soil moisture and surface temperature, changes in irrigation could also affect soil processes and exchanges of greenhouse gases and chemically reactive compounds between the surface and the atmosphere (Liu et al, 2008). Performing irrigation experiments on an Inner Mongolian Steppe, Liu et al. (2008) observed a significant sensitivity of the ecosystem $CO_2$ respiration to increased water input during the vegetation period, whereas the effects on $CH_4$ and $N_2O$ fluxes were much more moderate. In order to study the impact of irrigation on ozone and pollutants in the Central Valley of California, Li et al. (2016) implemented an irrigation method in the model WRF-Chem and showed an increase in surface primary pollutant concentrations within the irrigation zone. They also calculated an enhancement in the horizontal transport of ozone and other pollutants from irrigated to unirrigated areas near the ground surface. However, few studies have been published so far on this topic from a **biological or chemical perspective**, and the effect of irrigation on biological processes or on the atmospheric chemical composition therefore remains poorly quantified.

### Fertilization.

Since the Second World War, the use of synthetic N fertilizers largely increased, with half of the quantity ever used being applied in the last 20 years (Erisman et al., 2007). The growth of nitrogen fertilization

threaten water sources (e.g. eutrophication of surface waters, pollution of groundwater, acid rains), soils (e.g., soil acidification), climate via GHG emissions, and air quality.

Few studies investigated the impact of fertilizer use from a **physical perspective,** and yet physical interactions between the surface and the atmosphere could be affected. Based on a long-term experiment of fertilizer and amendment application running for 70 years, Pernes-Debuyser and Tessier (2004) observed that physical properties of plots were significantly affected, especially those related to soil-water relations. In spite of the preservation of their porosity, plots became more sensitive to the degradation of their hydraulic properties. Similarly, Hati et al. (2008) showed, in the case of an intensive conventional cultivation in sub-humid tropics in India (acidic Alfisols), the importance of soil management practices in maintaining the soil physical environment, with potential impact on soil aggregation, soil water retention, microporosity, available water capacity or bulk density.

From a **Biological** perspective: The additional source of nitrogen has different impacts on the atmosphere, mainly linked to an increase in reactive nitrogenous emissions ($NH_3$, $NO_x$) (Fowler et al., 2009; 2013; Galloway et al., 2003) but also in emissions of a GHG such as $N_2O$. Increase in production also affects leaf area index and plant height and therefore surface properties and physical exchanges with the atmosphere. Finally, fertilization also influences soil microbial characteristics and, consequently, exchanges of several gaseous compounds (Marschner et al., 2003; Cinnadurai et al., 2013; Joergensen et al., 2010; Murugan and Kumar, 2013). Grassland usually stores considerable amounts of carbon in the soils, mainly due to a permanent plant cover and to a relatively large belowground biomass (Bouwman, 1990; Casella and Soussana, 1997). However, the amount of stored carbon and the emission of greenhouse gases depend on the management of this grassland (ploughing, fertilization, pasture, etc.) (Soussana et al., 2004 ; Lal, 2004) and on climatic conditions (Hu et al., 2001). Some studies suggest that increased nitrogen fertilization can enhance C storage in grassland. On the other hand, nitrogen fertilization increases leaching and emissions of $N_2O$ and other nitrogen species (e.g., $NH_3$, $NO$) to the atmosphere, with negative consequences on air quality (Flechard et al., 2005, Senapati et al., 2014, Chabbi et al., 2015).

From a **Chemical** perspective: The increase in $NH_3$ emissions to the atmosphere can have a serious impact on air qualiy through the formation of secondary organic aerosols. Agricultural practices and techniques that reduce the evaporation of manure and urea and the use of N fertilizers help in lowering ammonia emissions from agriculture as documented in Europe, where 90% of the total ammonia emissions comes from agriculture (-9% over 1990-2002; Erisman et al., 2008). In China, where N fertilizer application rose by 271% over the 1977-2002 period, with an increase of 71% only in grain production (Ju et al., 2009), Ju et al. (2008) suggested to reduce by 30-60% N application rates. This agricultural management practice would still ensure crop yields and N balance in between rotations and would reduce economical costs for farmers, while substantially reducing N losses to the environment.

### Soil Surface conditions.

From a **Physical** perspective: Several crop management techniques (e.g., cover crops, double cropping, no-tillage) have a direct effect on regional climate through changes in surface-atmosphere fluxes and surface climate conditions, and are considered among geoengineering options. When tillage is suppressed, crop residues are left on the field, resulting in two counteracting mechanisms: albedo increases while evaporation reduces (Lobell et al., 2006; Davin et al., 2014; Wilhelm et al., 2015). Surface albedo increases by 10 % and lowers hot temperature values by about 2°C, however the effect on the mean climate is negligible. Climate effect of two growing seasons per year has been largely untested. Only Lobell et al. (2006) have shown via modelling that this experiment has a small impact on a temperature on multi-decadal time scales when compared to practices as irrigation. However, more recently Houspanossian et al. (2017) have observed through satellite imagery difference in reflected radiation between single and double-cropping up to 5 $W.m^2$. Similar to tillage/no-tillage mechanism, differences over South America were induced by a longer fallow period in the simple cropping case. Seed sowing dates also likely plays a role in surface energy balance, due to the modification of the growing season length (Sacks and Kucharik, 2011).

Among agriculture practices, as an alternative to biomass burning and natural decomposition, the use of charcoal from biomass pyrolysis to enrich soils may reduce $CO_2$ emissions. However, as side effect, the resulting darker soil increases the local radiative forcing through albedo change and offsets the sequestration effect up to 30 % according to Bozzi et al. (2015), who analysed based on observations of agricultural field albedo. Biochar has similar effects (Usowicz et al., 2016; Meyer et al. 2012).

From a **chemical** perspective: Fallow lands are potential sources of dust and coarse aerosols ($PM_{10}$), especially in regions where gusty winds dominate. Insufficient crop residues on the surface and finely divided soils by multiple tillage operations expose fallow land to wind erosion thus contributing to poor air quality (Lopez et al., 2000; Sharrat et al., 2007). In addition, wind erosion is likely to reduce crop yields by removing the richest fraction of soils, reducing the water-holding capacity of soils and enhancing soil degradation. Compared to conventional tillage (i.e., mouldboard ploughing followed by a compacting roller), alternative or reduced tillage practices (e.g., chisel ploughing) prevent wind erosion during fallow periods in semiarid Aragon (Lopez et al., 2007). In addition, reduced tillage improves soil protection by lowering wind erodible fraction of soil surface (-10%), increasing fraction of soil covered with crop residues and clods (+30%) and enhancing soil roughness (15% compared to 4% under conventional tillage). These agricultural practices therefore have the potential to modify aerosol sources by modifying the state of surfaces.

From a **biological** point of view, the conditions of the soil surface and the management of crop residues highly affect soil quality as well as the functioning and the abundance of soil microorganisms (Smith et al. 2015; 2016). In terms of exchange with the atmosphere, this results in soil structural changes affecting soil porosity directly influence the emissions of $NO_x$ and bCOVs (Gray et al. 2010; Bertram et al. 2005). Effects can also be seen on  soil organic matter content and degree and rate of decomposition therefore

affecting emissions of several nitrogen compounds therefore affecting GHG balance (emissions of $N_2O$ vs. Storage of carbon) (Longlong et al., 2018) and air quality ($NH_3$, $NO_x$ emissions) (de Ruijter et al. 2010). On the other hand, soil surface conditions also influence the deposition of $O_3$ (Stella et al. 2019) and potentially other highly reactive atmospheric compounds such as pesticides (Alletto et al. 2010).

**Fire.**

Fire is still largely used as a traditional agricultural practice (e.g., slash-and-burn agriculture, pest-control, promotion of the growth of fresh grass for grazing) and to convert forests to pasture/crop-lands, especially in tropical regions (Yevich and Logan, 2003). On a local scale, intensive mechanized grain agriculture reduces the use of fire. However, the wealth generated from intensive agriculture may be reinvested in traditional extensive land uses that promote fire (Wright et al., 2004).

Generally, fires can impact soil colour, pH, bulk density, soil texture, and therefore be critical for **physical** surface-atmosphere exchanges, together with **biological** properties of soil such as species richness and micro-organisms content (Thomas et al., 2014; Verma and Jayakumar, 2012; Savadogo et al., 2007). However the impact of fires from a physical or a biological perspective has been poorly investigated, especially regarding the long-term effect (Dooley and Treseder, 2012; Pressler et al. 2018). From a **chemical** perspective: Fire has impacts on both photochemical pollution ($O_3$ production) and aerosol loading. During fire episodes, $O_3$ production switches from a VOC-sensitive regime in nascent smoke plumes (i.e., first hours of burning and close to the ignition point) to a $NO_x$-sensitive regime as the plume ages. In nascent smoke plumes $NO_x$ levels are high and photochemical activity is low. Smoke plume aging decreases $NO_x$ levels via atmospheric dilution and chemical reactions, resulting in increased $O_3$ production (e.g., Jost et al., 2003; Trentmann et al., 2003; Yokelson et al., 2003; Mason et al., 2006; Singh et al., 2012). During fire episodes, $O_3$ levels may reach hazardous values, with the 8h-average $O_3$ concentration often exceeding air quality standards (around 50-75 ppbv; Bytnerowicz et al., 2010). Fires also release huge amounts of both coarse- and fine-mode aerosols, leading to concentrations that largely exceed background levels (Phuleria et al., 2005 Hu et al., 2008) and that substantially affect visibility (Val Martin et al., 2015). Over Singapore, Indonesian fires caused the average daily minimum horizontal visibility to reduce, firstly, to less than 2 km, and later to 500 m (Goldammer et al., 2009). Fire emissions encompass as well aerosol precursors such as $NH_3$ and BVOCs.

**Forest management.**

Forest management mainly relies on tree species selection, fertilization, litter raking, thinning and clear-cutting (Eriksson et al., 2007), together with planting and harvest types, burning and understory treatment.

From a **Physical** perspective: Along with crop management, forest management could provide similar impact for local climate but is still poorly investigated (Bellassen and Luyssaert, 2014; Luyssaert et al., 2014), although forested areas cover one third of the global land surface (Klein Goldewijk, 2001). The

large conversion of broadleaved to managed conifers forest resulted in biogeophysical changes which contributed to higher temperatures instead of attenuating them.

From a **biological** perspective: Through modelling, Naudts et al. (2016) showed that two and a half centuries of forest management in Europe may not have mitigated climate warming, contrary to what was sometimes assumed until now. With regard to atmospheric carbon budget, forests were altered from acting as a carbon sink to a carbon source, because of the removal of litter, dead wood, and soil carbon pools.

From a **chemical** perspective: by modifying the surface characteristics, forest management can change sources and sinks of reactive compounds, and therefore affect air quality. Conversely, forest management can also be a tool when targeting air pollution reduction. Using a coupled-model approach, Baumgardner et al. (2012) analysed the improvement of air quality by a forested peri-urban national park in the Mexico City megalopolis and underlined that their results can be used to understand the air quality regulation potentially provided by peri-urban forests as an ecosystem service, together with the regional dynamics of air pollution emissions from major urban areas.

### 3.3. Synthesis of Current knowledge

In the context of LULCCs and LMCs, the importance of land-atmosphere interactions for climate and air quality have been analysed in many studies published over the past two-three decades, exploring a large range of scales. We summarize here the current state of knowledge emerging from the articles we reviewed. For each of the LULCC category (land cover change / agricultural intensification/ urbanization) considered in this article, the direct and cascading effects on the physical, biological and chemical processes are synthesized in Table 2.

Regarding physical processes, the works published so far on deforestation / afforestation mainly apply a modelling approach where the different processes involved (surface albedo, radiation and energy budget, etc.) are overall well understood. These works compare the effects between current and pre-industrial (potential) vegetation, representative of a time-period with few (no) human activities. There is no single/simple response to these LULCCs as the sign and amplitude of the effects on temperature and precipitation depend on the latitude, on the pre/post vegetation types, and the landscape configuration. The effect on air temperature remains mainly unclear in most temperate regions, as this is where changes in the radiative budget compete with changes in the hydrological cycle. Regarding wetland drainage for agricultural purposes, very few studies investigate its impact on local/regional climate, in spite of the size of the areas affected. Via a modelling approach, existing studies show contrasting effects of wetland drainage on daily temperatures. Among agricultural management practices, irrigation is largely used all over the world and its impact on climate has been discussed in several studies using both observations and modelling. These works analyse both the greenhouse and the transpiration effect of irrigation, and suggest that the local cooling of irrigation might have partly

masked the 20th century climate warming at regional scales. The potential impacts on local to regional climate of other agricultural management practices, such as field preparation for planting, charcoal use for soil enrichment or forest management, remain poorly investigated but existing studies suggest their impacts on specific seasons and on climate extremes may be significant. UHI effect on climate is largely

analysed in the literature, and the reasons for a warmer climate are explained by a change in the surface radiative budget, a less efficient energy dissipation due to less convection, and heat release by human activities. However, the overall impact significantly varies depending on the time of the day, season, human activities, geographical location and spatial organization of the urban fabric. Moreover, almost no study refers to realistic landscapes and realistic changes, with potential compensation or amplifying

effects. This is a challenge ahead as existing studies may not yet provide enough information to anticipate the impacts of realistic land use scenarios.

Whatever the land change described above, there are many numerical evidences that its effect on extreme weather/climate events is quite larger than their impact on mean seasonal or annual climate. Focusing on central France, for example, Stéfanon et al. (2014) demonstrated that if this part of France

had been partially afforested in 2003, the June heatwave would have been aggravated by up to +3°C, while the August one would have been dampened by as much as -1.6°C locally.

Enhanced extreme winter cold temperatures and lengthening of frosts have also been identified by Marshall et al. (2004a) in response to the drainage of wetlands and replacement by agriculture in Florida. By altering extreme conditions rather than the mean regional climate, these LULCCs have been

responsible of reduced crop yields in the region.

Pitman et al. (2012a) carefully carried out a multi-model analysis at the global scale of the impacts of historical land cover changes on extreme temperature and precipitation indexes (using the indices recommended by the CCl/CLIVAR/JCOMM Expert Team on Climate Change Detection and Indices, ETCCDI, based on daily maximum and minimum temperature and daily precipitation),. They found

that, wherever the land-cover change induced a decrease (resp. increases) in averaged temperature, the extreme temperatures were also reduced (resp. increased). By comparing the LULCC-induced changes to those resulting from the increase in atmospheric $CO_2$ and sea-surface temperatures during the same historical period, the authors found that the LULCC-induced changes may be as large as changes triggered by global warming, sometimes even larger, and potentially of opposite sign.

Any land cover conversion or land management that favors the increase in (or reversely the decrease in) evapotranspiration during a specific season (e.g., irrigation, crop intensification versus e.g. deforestation, tillage suppression) has consequences on extreme daily temperatures, without affecting the mean seasonal temperatures. LULCCs and LMCs generally decrease maximum temperatures ($T_{max}$) and thereby reduce the diurnal thermal amplitude (Davin et al. 2014; Thiery et al. 2017).

Focusing on biological processes, several studies show that, via changes in temperature and soil moisture, deforestation affects nutrient mineralization in soils by enhancing carbon and nitrogen release to the atmosphere and the environment. Some of the released gases have a significant warming potential

(CO$_2$, CH$_4$, N$_2$O, for instance) or they are involved in the ozone cycle or aerosol formation (e.g., emissions from fire clearing). These compounds can affect the climate at local, regional or global scales. Several studies show that peatland and wetland conversion affect climate from the local, through e.g. evaporation and surface temperature change, to the global scale, by e.g. changing surface emissions of greenhouse gases such as CO$_2$ or CH$_4$. Impacts of agricultural management on climate and air quality are widely investigated via modelling or experimental studies at the local scale; however, very few studies investigate the impact of agricultural management on climate through changes in biological processes at the landscape, regional or global scales. Land-use intensification and fertilization are shown to have the potential to affect climate, through modification of greenhouse gas emissions and carbon sequestration, but also on regional air quality via the emission of different reactive species such as NH$_3$, NO$_x$ and several VOC species. This highlights the complex interactions and feedbacks between chemistry and biology, such as the interactions between ozone and reactive nitrogen in the context of their mutual impacts on ecosystems. Key results showed that exposure to ambient O$_3$ concentrations was reducing the Nitrogen Use Efficiency of plants, both decreasing agricultural production and posing an increased risk of other forms of nitrogen pollution, such as nitrate leaching (NO$_3$). Ambient levels of aerosols were also demonstrated to reduce the ability of plants to conserve water under drought conditions. These results clearly show the tight interactions between the atmospheric chemical composition and the ecosystem and agroecosystem functioning, with a strong need for further model adaptation and investigations.

At last, by affecting surface emissions and atmospheric chemical processes, LULCCs and LMCs have the potential to affect air quality, by changing air pollutant concentrations, and the local-to-global climate, by modifying greenhouse gases (O$_3$, CH$_4$, CO$_2$, etc.) or levels of radiative compounds (e.g., aerosols). Most of the studies published so far apply a modelling approach and analyse the impact of regional- or large-scale changes in land-cover on land-atmosphere chemical interactions (deforestation in tropical areas, preindustrial to present-day or future changes in vegetation distribution, etc.). The increase in biofuel and oil palm plantations for energy and food production has been targeted by several studies. Among the different agricultural practices, fertilization, agriculture fires and fallow periods have been shown to affect air quality by emitting ammonia, ozone precursors and/or aerosols. However, the impact of land and agricultural management on air quality, and potentially climate, through changes of land-atmosphere chemical interactions, remains poorly investigated. An increasing number of studies assessed the impact of urbanization on land-atmosphere chemical interactions and air quality, with a raising interest on the impact of urban trees on ozone pollution, through changes in BVOC emissions.

### 4. Interactions between different land cover, uses and managements over a mosaic landscape: impacts on land-surface exchanges

In the real world complexity arises where territories are composed of a mosaic of very diverse landscapes in which physical, biological and chemical processes take place and interact altogether. Areas of agricultural surfaces, covered by different types of crops and cattle, forests composed of a varying

mixture of plant types, urban and peri-urban areas of different sizes co-exist next to each other, sharing one single atmosphere with no boundaries. Therefore, one homogeneous parcel has the potential to influence surrounding ones, over a range of time and geographic scales that will depend on considered processes, as illustrated in Figure 4. Horizontal transport of air masses promotes water, heat or pollutant exchanges between surrounding areas. Regarding air quality, compounds emitted from one area can be transported to remote places, depending on their lifetime, undergo chemical transformations in the atmosphere, and consequently influence the chemical composition of the air in distant regions. In this section, we will draw an overview of possible interactions between physical, biological and chemical processes, and we will analyse them over a mosaic of landscapes from three different perspectives: local climate, air quality and ecosystem functioning. These changes and interactions ultimately modify local climate and air pollution as specified in Sect. 3.

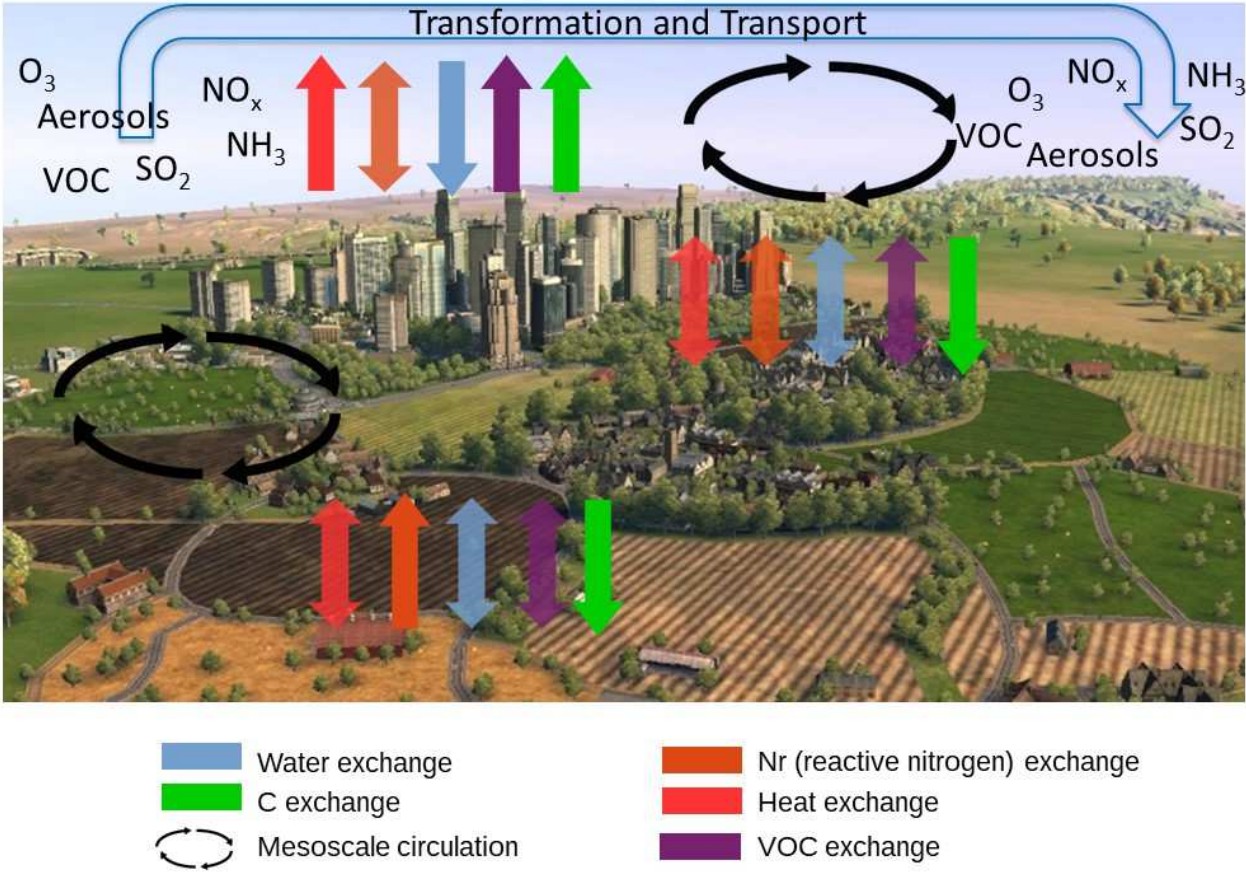

**Figure 4: Interactions between different land-uses and major trend of gaseous flux direction from each land-use type. Different colours represent different scalars. Mono-directional arrows indicate where scalars are mostly emitted or deposited by the land-use. Bi-directional arrows indicate where scalars can be both emitted or deposited depending on atmospheric and ecosystem conditions.**

### 4.1. Local- to Meso-climate perspective

Horizontal advection from one LULC to another can significantly modify local climate downwind. For instance, urban areas not only heat their local environment but also their surroundings due to horizontal transport of warm air masses to suburban and rural environments. As reported by Bohnenstengel et al.

(2011), suburban areas downwind London are 1°C warmer during night-time than upwind ones due to heat advected from the city center. Similarly, Heaviside et al. (2015) found that temperatures downwind of Birmingham were up to 2.5°C warmer than those upwind during the heatwave of August 2003. Sarrat et al. (2006) found that temperatures in suburban area were 1.5 °C warmer when including UHI effect in their simulation than without considering it. They also highlighted that UHI is displaced to suburban areas by horizontal advection and forms an urban heat plume. This effect can extend to about tens of kilometers downwind (Brandsma et al., 2003; Bohnenstengel et al., 2011). However, this issue is closely linked with wind speed (Kim and Baik, 2002; Brandsma et al., 2003): a minimum wind speed (> 0.1 m/s) is required for urban heat advection to become effective, while for larger wind speeds (> 5 m/s) the mixing of the heat plume with the overlying atmosphere decreases this effect (Brandsma et al., 2003). Moreover, spatial heterogeneities induced by LULCCs are likely to produce atmospheric circulations - similar to the sea/lake breeze (so-called non classical mesoscale circulations)- or to modify the magnitude of pre-existing background wind, as documented experimentally (Briggs, 1988; Mahrt et al., 1994) as well as numerically (Mahfouf et al., 1987; Hadfield et al., 1992; Shen and Leclerc, 1995; Avissar and Schmidt, 1998; Stohlgren et al., 1998). Heterogeneities of surface properties and heat fluxes over contrasting areas are the main and required criteria for this mesoscale process (Anthes, 1984; Segal et al., 1988) that can generate over bare soil-vegetated areas, irrigated-unirrigated regions, urban-rural areas, or mountain-valley structures (Avissar and Pielke, 1989). Distribution of heating at scales of the order of tens of kilometres is necessary to initiate such circulations (André et al., 1990; Mahrt and Ek, 1993; Segal and Arrit, 1992; Wang et al., 2000). The generation of mesoscale circulations carries heat and water vapour which have a significant influence on the planetary boundary layer dynamics and properties (temperature, water vapor, cloudiness and vertical heat flux) (Anthes, 1984; Segal et al., 1988; Avissar and Liu, 1996; Avissar and Schmidt, 1998). For instance, deforestation upwind of montane forests results in warmer and drier air, which induces thinner clouds and a reduction in air humidity (Nair et al., 2003; Ray et al., 2006). Conversely, downwind of heavily irrigated areas, a rainfall increase of 15-30% was observed over the U.S Great Plains (DeAngelis et al., 2010). Finally, although it is clear urban areas alter rainfall events in their surrounding (Shepherd, 2005), it is difficult to assess precisely the localization and magnitude of induced rainfall events. For instance, Shepherd et al. (2002) reported that the maximum rainfall rates were between 48% and 116% larger downwind the city than upwind, while Dou et al. (2015) found that minimum rainfall occurred directly downwind the urban area (up to -35%), whereas maximum values along its downwind lateral edges.

### 4.2. Ecosystem functioning perspective

It has long been acknowledged that simultaneous interactions exist between landscape organization, structure, and biological functioning. Human activity also plays a major role in regulating and shaping those dynamic biogeophysical interactions at the landscape level. Organisms not only respond to their physical environment, but also they directly modify and control their physical environment in ways that

promote their own persistence. Several scientific disciplines such as 'ecological stoichiometry' (Sterner and Elser, 2002), 'ecosystem engineering' (Jones et al., 1994), and 'biodiversity and ecosystem functioning' (Loreau et al., 2002) illustrate how living organisms shape their own environment through the biogeochemical alteration in a multi-dimensional environment. These different interactions between

5 animals, vegetation and physical and chemical processes can be illustrated through different examples such as alteration of soils and water quality, seed and spore dispersal, and competition for soil, moisture and light (Hastings, 2004).

There are several examples in non-anthropized environments, which show the feedbacks between macro-fauna, vegetation, soil formation, sediment transport and ultimately landscape formation. For

example, Van Hulzen et al. (2007) demonstrated how certain plant species both modify their habitat via their own physical structures, and respond to those modifications. The plant modifies its environment so that it becomes more locally favourable. However, these modifications create small 'islands', therefore limiting the plant to spread. There is a consensus that climate-driven changes in precipitations will influence the pattern and vegetation type (and animals) in landscapes, which will in turn influence

physical processes. However, today, human activity mainly shapes the landscape we live in. For example, high inputs of fertilizers and pesticides degrade the habitat quality, while the expansion of arable lands promotes widespread landscape homogenization (Robinson and Sutherland, 2002). Studies over the last two decades have emphasized the importance of landscape scale effects in these processes (Benton et al., 2003; Hole et al., 2005; Matson et al., 1997; Swift et al., 2004; Vandermeer et al., 1998).

Biological processes respond differently based on landscape structure. For example, Vinatier et al. (2012) showed that pest dispersal may be of greater importance in fragmented rather than homogenous landscapes. By considering the link between ecological processes and landscape composition, one can therefore evaluate the impact of habitat loss and fragmentation due to human activity on different population dynamics (Wiegand et al., 1999, 2005; Fahrig, 2003). In this context, some results are

sometimes contradictory. Roschewitz et al. (2005) and Thies et al. (2005) found that complex landscapes, characterized by a higher proportion of semi-natural habitats, increase aphid parasitism rate but also aphid abundances. On the other hand, studies by Caballero-López et al. (2012), Costamagna et al. (2004), Menalled et al., 2003 and Vollhardt et al. (2008) showed that landscape complexity has no effect on parasite diversity.

Another critical issue linked to ecosystem functioning and landscape structure is soil quality. Montgomery (2007) showed that conventionally ploughed fields generally erode at rates typical of alpine terrain under native vegetation. However, LULCCs is not only the cause but can also be the consequence of erosion processes (Bakker et al., 2005). Landscape alteration also influences nitrogen availability through its impact on organic matter through fire (Mataix-Solera et al., 2011; Debano and

Conrad, 1978), tree-fall (Schroth et al., 2002; Mladenoff, 1987; Vitousek and Denslow, 1986; Muscolo et al., 2014; Feldpausch et al., 2011) and forest practice (Fujisaki et al., 2015; Guimarães et al., 2013; Berenguer et al., 2014; Bormann and Likens, 1979; Vitousek and Matson, 1985), which all produce

patchy landscapes. Soil nitrogen alterations can have important immediate consequences on N cycling as volatilization, recycling of organic matter from aboveground biomass, reduced uptake by plants, altered rates of solution transport through the soil profile, and elevated mineralization. These disturbances can indirectly affect the ways in which different species colonize disturbed areas and

recycle N. Over longer periods, the species composition resulting from disturbance might affect nutrient supply and influence total carbon and N pools, element ratios, and pH (Zinke, 1962; Wagle and Kitchen, 1972; Christensen and Muller, 1975; Christensen, 1977; Raison, 1979; Boerner, 1982).

Proximity of a natural ecosystem to an urban area also alters this ecosystem functioning as it has been shown through several studies. As mentioned above, air quality and more precisely ozone concentrations

affect leaf photosynthesis and therefore ecosystem production. The degradation of ecosystems in proximity with big cities has been studied mainly in the perspective of analysing the effect on ecosystem services and the subsequent effects on populations in general and vulnerable populations in particular (Elmqvist et al., 2013, Haase et al., 2014).

### 4.3. Air quality perspective

As illustrated in Sect. 3, LULCCs and LMCs directly influence the local air pollution via changes in the intensity and variability (temporal and geographical) of chemical emissions (e.g., BVOCs from tree species, $NO_x$ emissions from soils and fertilization) or in chemical processes and regimes (e.g., from $NO_x$- to VOC-sensitive regimes in $O_3$ production). In addition, by modifying land-atmosphere interactions, LULCCs and LMCs can indirectly affect air quality by altering atmospheric circulation

(i.e., vertical mixing and advection) with consequences on the dispersion of pollutants and of pollutant precursors.

Pollutant dispersion in the planetary boundary layer (PBL) is strongly influenced by changes in the PBL height and in convective transport, which are triggered in turn by modified land-atmosphere energy transfer (Ganzeveld and Lelieveld, 2004; Civerolo et al., 2007, Rendón et al., 2014; Wagner and Schäfer,

2017). Intense convection makes the PBL deeper; this condition, together with enhanced advection, increases pollutant dispersion. In the troposphere, $O_3$ and secondary aerosol production depends on the abundance of their precursors (i.e., $NO_x$ and VOCs). Increased dispersion may reduce concentrations of precursors, finally reducing ozone production. On the contrary, stagnant atmospheric conditions often associated with low advection and strong thermal inversion, limit pollutant dispersion and favour $O_3$

production.

Stagnant atmospheric conditions correspond to low winds, intense solar radiation and high surface temperatures. Under these sunny and warm conditions, $O_3$ production increases because of the direct effect of altered radical production and photochemistry (Fiore et al., 2012) and the indirect effect of enhanced BVOC emissions (e.g., Cardelino and Chameides, 1990; Taha, 1996; Val Martin et al., 2015).

At the urban scale, Cardelino and Chameides (1990) estimated a rise of 25% in BVOC emissions due to warmer temperatures, in spite of a decrease of forest areas by 20%, due to growing urbanization. In

terms of $O_3$ production, increased BVOC emissions ruled out the benefits of a substantial reduction in anthropogenic VOC emissions (-50%) via "clean air" policies. Enhanced BVOC emissions may feed as well the aerosol loading via BSOA production (e.g., Cardelino and Chameides, 1990; Nowak et al., 2000). The influence of atmospheric conditions on the aerosol loading depends on the aerosol type.

Nitrates dominate under cold temperatures, while sulphates prefer warm temperatures. Hygroscopic aerosols benefit from high humidity. For most aerosols, the precipitation rate directly controls the aerosol loading since scavenging (wet deposition) is the main sink for aerosols.

Surface roughness and vegetation conditions (i.e., plant type, plant health, heat stress) strongly affect both aerosol and $O_3$ dry deposition. Ozone deposition involves biological processes and decreases with decreasing surface and leaf wetness (Klemm and Mangold, 2001). When vegetation is not water-limited, ozone can be absorbed by leaves via stomatal uptake. Above a certain threshold, $O_3$ deposition reduces photosynthesis, plant growth, biomass accumulation and crop yields, and affects stomatal control over plant evapotranspiration (Ainsworth et al., 2012). Hence, although $O_3$ deposition by stomatal uptake improves air quality, it may result in plant damage in the long term. Ozone deposition depends as well on mechanical processes. By increasing surface roughness, trees reduce horizontal wind speeds and limit pollutant dispersion leading to increased ozone levels both locally and regionally (e.g., Nowak et al., 2000). On the other hand, reforestation of croplands (Trail et al., 2015) or vegetation increase in urban areas (Taha, 1996) improve $O_3$ deposition and reduce $O_3$ concentration. This ozone-reducing mechanism combines with other afforestation-driven effects, such as reduced $NO_x$ emissions from soils and fertilization and lower surface temperatures, and competes with higher BVOC emissions from trees, which may trigger $O_3$ production (Trail et al., 2015). Ecosystem distribution can also be a significant driver of deposition efficiency, which is still not well quantified. A shift from croplands to grasslands reduces dry deposition velocity and increases ozone concentration (Val Martin et al., 2015). Taking into account the 2050 RCP 8.5 vegetation distribution, which is characterized by an expansion of land used for crops and pastures at the expense of forests, Verbeke et al. (2015) calculated a rise in the surface ozone deposition velocity, relative to the present-day values, up to 7 % in tropical Africa and up to +18% in Australia. Moreover, although pollutant deposition on trees significantly reduces ozone levels, this effect is hampered as the PBL height increases (Nowak et al., 2000). On the contrary, a conversion from forests to croplands modifies stomatal activity and affects deposition rates of trace gases, such as ozone, more than changes in LAI (Trail et al., 2015). Furthermore, for aerosols, conversion from forests to croplands reduces aerosol dry deposition because of decreased surface roughness. In cities, promoting green infrastructures have been considered as a tool to improve air quality, but their actual impact on the atmospheric chemical composition is only quantified in a few studies (Churkina et al., 2017; Ren et al., 2017). A recent review by Abhijith et al. (2017) shows that the choice of infrastructure is critical, with for instance low-level green infrastructure (hedges) improving air quality compared to high vegetation canopies.

To summarize, LULCCs and LMCs affect air quality directly, by influencing the sources and sinks of reactive compounds at the surface, and indirectly, by modifying environmental conditions (temperature, mixing) in which surface-atmosphere chemical exchanges occur. By modifying the air chemical composition and possibly affecting the occurrence of pollution episodes, changes described so far have the potential to affect, in turn, vegetation distribution and growth. Consequently, these changes could also affect retroactively physical and biological processes involved, with potential impact on meteorological conditions and climate, at the local and regional scales. To investigate future air quality, future LULCCs and LMCs should be accounted for in meteorological models that provide forcing to chemical-transport models. If not, projections of future air quality will not account for the indirect influence of land-atmosphere interactions on the evolution of air quality (Civerolo et al., 2000).

## 5. Futur Research

In Section 3 we have reviewed recent progress, both from an experimental and modelling point of view, in our understanding of processes and mechanisms involved in land-atmosphere interactions at different scales, going from organ to plant, from plot up to regional scales. In Section 4 we have discussed studies focusing on the interactions between the different landscape structures that affect local climate and air quality. Through these analyses, we have highlighted that the representation of interactions and feedbacks between the different compartments (physics, biology, chemistry) and surfaces (urban, peri-urban, agricultural, natural, etc.) is crucial when investigating the impact of LULCCs on climate from small to larger scales. Based on these analyses, in the present section we identify actual knowledge gaps in the processes, feedbacks, methodologies and parameterizations currently used to reproduce interactions between land, LULCCs and the atmosphere. We summarize below the limitations that exist today and that restrain our capacity to investigate the effects of LULCCs and LMCs on local climate and air quality at different scales using a modelling and/or an experimental approach, while considering all the interactions involved.

### 5.1. Challenges ahead

**The first challenge** is the **lack of integration between the different known processes.** It is not easy to design an experimental protocol that allows us to differentiate between the impacts relative to each different process (Pitman et al. (2012b). Although several initiatives are being conducted to couple model and ecocystem based experiments to allow disentangling of processes and better model performance (ex. Norby et al. 2015; Medlyn et al. 2015) it is still a big challenge today (Higgins 2017). Nearby urban areas, for example, strong pollution levels -with especially high ozone concentration- may directly affect plant productivity through atmospheric advection of those pollutants downwind from the city. In such a case, surface and air temperature may be perturbed in rural regions through changes in vegetation characteristics (e.g., stomatal opening, albedo) and fluxes (e.g., latent heat flux). A coupled land-atmosphere model that does not account for chemistry processes will therefore not be able to correctly reproduce surface climate and vegetation status in the rural environment. In addition, the

representation of urban areas is often very simplified. For instance, regarding atmospheric chemistry, emission sources are usually prescribed, which do not allow accounting for feedbacks. Hence, a coupled urban – vegetation – chemistry model is a necessary development, as also pointed out by Baklanov et al. (2014) in their review of online modelling of atmospheric and chemical conditions (i.e., online modelling refers to the numerical technique of having atmospheric and chemical conditions evolve in parallel with the atmospheric and the chemical modules exchanging information in the two ways at each time step).

Figure 5a illustrates the interactions between the different variables and processes involved in biosphere-atmosphere exchanges as discussed in the previous sections (that are not exhaustive with respect to the existing literature). Today most of these interactions (solid lines) are relatively well known but are not yet experimentally measured or jointly accounted for in regional global climate models, which we are targeting here. Whereas global climate models, such as those used for the "Coupled Model Intercomparison Project" (CMIP) exercises for the Intergovernmental Panel on Climate Change (IPCC), are now referred to as Earth System Models (ESM) that include a large spectrum of physical, chemical and biological processes in the modules that describe the atmosphere, biosphere and hydrosphere reservoirs, regional climate models have recently started to move towards the frontiers of regional ESM (e.g., Sitz et al., 2017).

**The second challenge** relies on the **detailed representation of the variety of surfaces** in the above-mentioned models. Indeed, surfaces such as cities, managed forests, mixed areas, wetlands or the variety of agricultural crops are either over-simplified (e.g., no distinction of forest species in a forest biome), or miss-represented (e.g., crops represented as a super-grassland), or absent (e.g., absence of wetland representation). Such gaps could be potentially bridged by using more sophisticated dynamic global vegetation models (DGVMs) than those currently used in climate models. In their analysis of DGVMs, Scheiter et al. (2013) pinpointed some of the limits of the current generation of DVGMs such as, for instance, the use of bioclimatic limits to force the modelled vegetation type to grow under the "correct climate" (the one that will guarantee the selected vegetation type to grow), or the parametrization of the number of species and the degree of functional diversity that is necessary sustain ecosystem function. The authors tested in a trait- and individual-based vegetation model some of the new concepts that could fit in the next generation of DVGMs (e.g., assembly theory and coexistence theory. Moreover, DGVMs could be coupled to chemistry models to gain a better description of the land surface as well as of the land management practice If such DGVMs may include the impact changes in air quality have on the functioning of the ecosystems they model, the reverse is not true. Most chemistry and transport models, for example, consider prescribed and fixed information for vegetation (distribution, areas, related characteristics such as leaf area index, stomatal resistance, etc.) and as well for land management and farming practices, which are relatively scarce at the regional and global scales. As this information is used to calculate emissions and deposition, it can strongly affect the assessment of atmospheric chemical

composition. Therefore, the numerical coupling between atmospheric chemistry and the terrestrial biosphere, or at least a more dynamic representation of vegetation in chemistry-transport models (Baklanov et al., 2014), is a crucial step forward the development of integrated numerical tools. Coarse-resolution models (e.g., global-scale, ~100 km) may be inadequate in separating different chemical

regimes that are triggered by emission patterns of biogenic and anthropogenic sources. However, nowadays, the integration of such loops in numerical models is limited because the various components of these interactions are developed by independent groups, in diverse surface models that are not all coupled to atmospheric models. This is of high importance, especially in short or long-term conditions where LULCCs and climate are meant to change significantly under the influence of human activities.

For instance, the variety of plant species encompassed in BVOC emission database is limited (e.g., Ashworth et al., 2012), with therefore incomplete information regarding emission geographical variability. This biases both the ability to describe and to properly evaluate BVOC emissions in modelling tools. Green roofs in urban-atmosphere models are generally represented through uniform, idealized, vegetation, while ecological papers have shown a large variability in the vegetation response

to climate, depending on species. Not accounting for such bio-diversity may affect the ability to calculate the exact cooling effect of those roofs. Moreover, studies often target emissions from a single sector (e.g., oil palm industry, biofuel production) without taking into account emission evolution in other sectors (other than oil crop/biofuel industry) or in nearby regions (e.g., Hewitt et al., 2009). The exclusion of emission sources other than those from LULCCs and LMCs may affect results (over- or

under-estimate) regarding ozone and aerosol levels. For example, most large-scale modelling studies use global vegetation models to investigate the interactions between the chemistry and the biosphere and adopt a simplified representation of ecosystems as a selection of plant functional types (PFTs). The PFT approach lumps individual plants with similar ecological characteristics and behaviours under the same vegetation type. Although the PFT approach works at the global scale, once applied at the regional

scale it may restrain the model skills in representing the ecosystem variability as well as the land management scenarios, which are often not accounted in the models, as also pointed out by Scheiter et al. (2013).

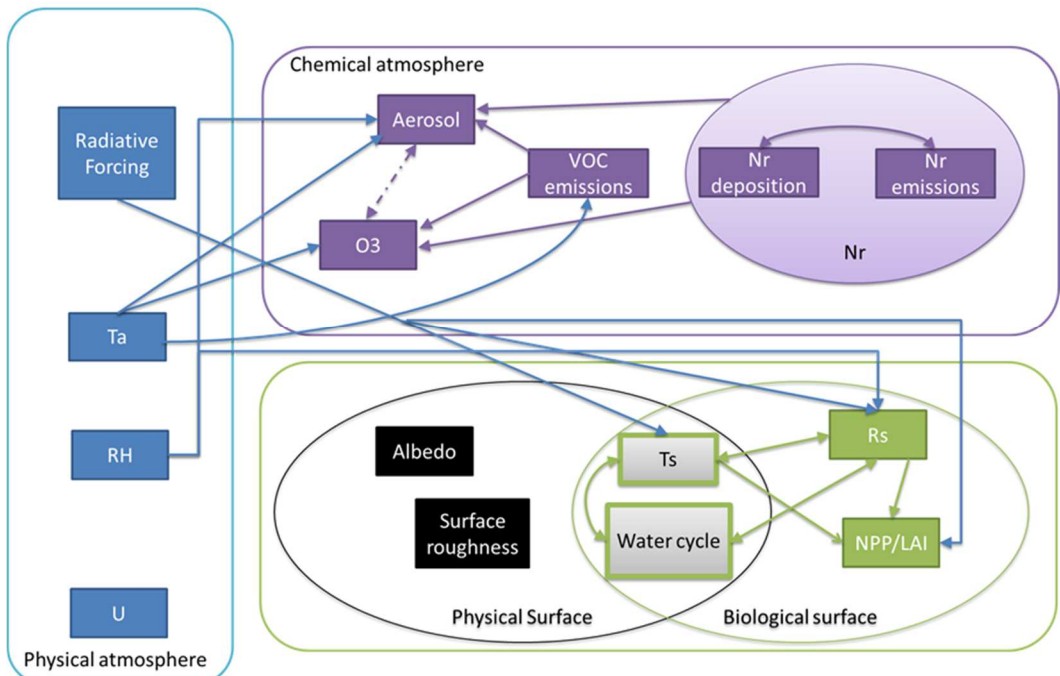

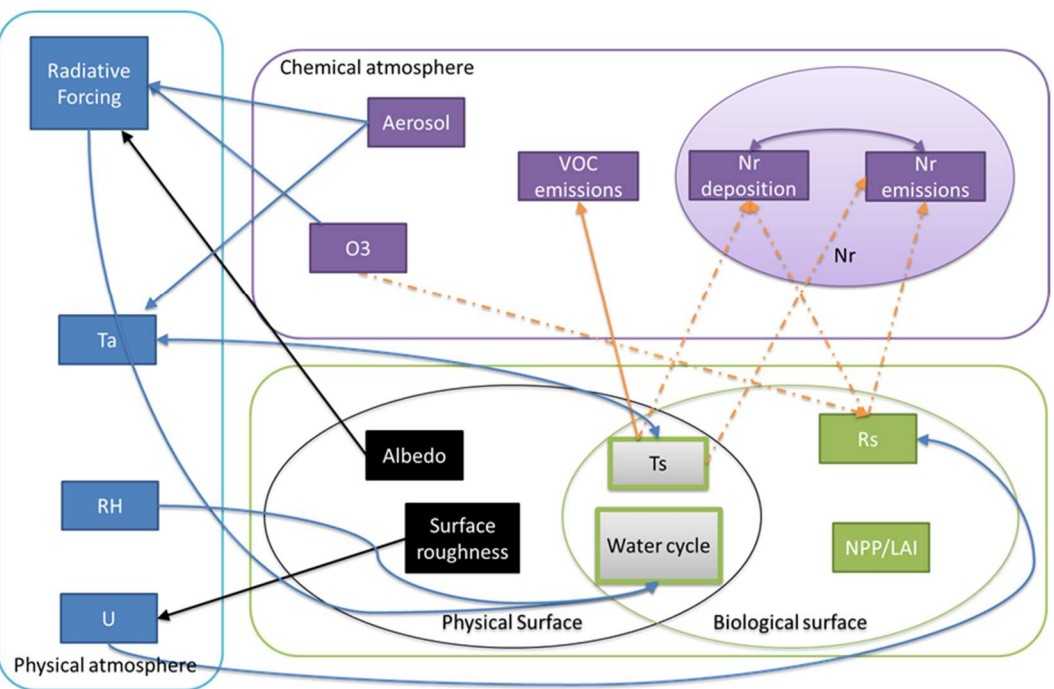

**Figure 5: Interactions between the different variables and processes (a) concerned in biosphere-atmosphere exchanges as well as feedbacks (b) involved between the physical and biological surfaces of an ecosystem and the physical and chemical compartments of the adjacent atmosphere. Full arrows represent well-documented processes and feedbacks, dashed arrows represent mechanisms having knowledge gaps or badly represented in most models. $N_r$ stands for reactive Nitrogen species, $T_a$ for air temperature, RH for air relative humidity, U for average wind speed, VOC for volatile organic compounds, $T_s$ for surface temperature, $R_s$ for stomatal resistance, NPP for net primary production and LAI for Leaf Area Index. This schematic covers most atmospheric variables discussed in the paper, but not all atmospheric variables that can be affected. Rainfall and cloudiness for example are amongst the ones that have been shown to be sensitive to land and are not discussed herein.**

**The third challenge** is the **need for observational data covering more temporal and spatial scales**. For example, various observations of BVOC or reactive N compound emissions have been published for European and North American ecosystems, while few observational studies target southern regions. Due to the absence of such dataset, it is complicated to perform robust evaluation of models at the adapted scales, as also pointed out by Arneth et al. (2008). There is obviously a missing link between the regional scale, at which most chemistry and transport models are run and local scales, where observations are collected. Such investigations could also help to improve parameterizations generally used in models. The dependency of certain processes to different plant species and pedo-climatic regions is indeed generally not well described in model parameterisations.

Lastly, some processes are known but are not yet implemented in models. Figure 5b represents the existing known feedbacks between the different compartments. The feedbacks between the biosphere and the atmosphere via the impacts of vegetation on chemistry (dashed lines) are an example of missing processes in the majority of models. For example, bidirectional exchange of reactive N compounds is well known today but few chemistry and transport models fully integrate N exchanges although some advances have been made concerning ammonia (Bash et al. 2013; Zhu et al. 2015). However, we are still missing process-level knowledge on some of those interactions.

### 5.2. Towards interdisciplinary approaches

This review has highlighted the need to connect different scientific disciplines (e.g., physics, ecology, biology, agronomy, chemistry) in order to correctly represent the impacts of LULCCs and LMCs on climate at various spatial scales. In the following, we illustrate the need for such connections using two examples of current challenges in Europe.

#### 5.2.1. Urban – agricultural – natural triptych in a N pollution context

While agriculture has been criticized for several decades for its impacts on water quality (nitrate and pesticides) and for its contribution to climate change (emissions of nitrous oxide and methane), the question of its contribution to air pollution in urban and peri-urban areas has emerged only recently in the public debate, with a particular resurgence in recent spring episodes of aerosol pollution. Ammonia, which is largely emitted by animal excreta and by the application of mineral and organic fertilizers, contributes to the formation of secondary aerosols. Hence, the reduction of its emissions is an important stake for the improvement of air quality. In recent years, control of ammonia emissions has become a major concern at regional, national and international levels and, since the end of the 1990s, a set of regulations has been put in place. To further reduce ammonia emissions, improve air quality and optimize costs and benefits requires a better knowledge and quantification of ammonia sources and as well an analysis of long-term strategies. France regularly undergoes peaks of aerosol pollution (PM10-PM2.5) especially at the end of winter-early spring, when favourable weather conditions coincide with the beginning of fertilizer spreading. In March 2014, high PM2.5 concentrations were observed in the

Paris Region, led to the introduction of alternating traffic, and therefore made citizens particularly aware of the issues of air quality. Predicting air quality at the regional level is crucial to understand these episodes and to recommend appropriate levers of action in the short term to limit the magnitude of these episodes. Air pollution not only affects human health, but also the overall productivity of ecosystems and crop yields, through increased dry deposition of N compounds and $O_3$, which in turn could affect BVOC emissions. In addition, by modifying plant functioning in terms of evapotranspiration and soil moisture status, ozone deposition may affect the hydrological cycle, which in turn will affect surface but also wet deposition of pollutants and nutrients.

We have here a typical example where scientists involved in agronomy, physics, biology and chemistry should interact to improve predictions of ammonia emissions, transport and reactions related to weather conditions, soil biological processes and plant phenology, to estimate feedbacks of air pollution on the functioning of involved ecosystems. However, to solve the problem, cooperation between farmers, urban planners and decision makers is required to define optimal fertilization dates and a territorial planning of urban and peri-urban areas that accounts for the distribution of agricultural activities around the city.

### 5.2.2. Urban greening – UHI - and impact on VOC / NOx / O₃ loop

Many studies have explored techniques to counterbalance the deleterious effects of urbanization on the local environment. Among the numerous solutions already proposed, urban greening is one of the most interesting since it could allow (i) an attenuation of the UHI (e.g., Shashua-Bar and Hoffman., 2000; Alexandri and Jones, 2008; Feyisa et al., 2014), (ii) a direct mitigation of air pollution via the absorption of pollutants by plants (Hill, 1971), and (iii) an indirect improvement of air quality through UHI mitigation since temperature partly drives and controls pollutant emission, dispersion, and formation (Sini et al., 1996; Kim and Baik, 1999; Stathopoulou et al., 2008).

On the one hand, green surfaces such as parks, gardens, or green roofs and walls contribute to mitigate the UHI and currently receive strong attention from both scientists and urban planners (e.g., Shashua-Bar and Hoffman, 2000; Akbari et al., 2001; Kumar and Kaushik, 2005; Alexandri and Jones, 2008; Feyisa et al., 2014) with some interdisciplinary and inter-community experiences already established (e.g. the Urban Climate Change Research Network, ; the MAPUCE project in Toulouse, , local projects in Stuttgart, New York). On the other hand, a growing number of studies focuses on urban air quality assessment to quantify impacts of urban vegetation (e.g., Yang et al., 2005; Novak et al., 2006; Escobedo et al., 2011; Selmi et al., 2016). Changes in planted species and their surfaces can indeed significantly impact the amount and fate of reactive compounds emitted, such as biogenic VOCs or nitrogen compounds, and therefore affect the air chemical composition in terms of gases and aerosols (Ghirardo et al., 2016; Janhäll, 2015, Taha et al., 2015). Nevertheless, feedbacks on air quality by UHI mitigation are not accounted for but could lead to air quality degradation, by affecting pollutant and especially ozone precursor dispersion (Lai and Cheng, 2009). To quantify to which extent urban greening can help

to mitigate urban local climate and atmospheric pollution, and its subsequent effects at the regional scale, it is therefore necessary to adopt interdisciplinary approaches (Baró et al., 2014), involving atmospheric physics and chemistry, but also urban planners. Indeed, although the role of urban form, urban fabric, and building arrangement and orientation on UHI mitigation was explored in previous studies (Stone and Norman, 2006; Emmanuel and Fernando, 2007; Shahmohamadi et al., 2010; Middel et al., 2014), it was not the case for atmospheric composition.

### 5.3. Bridge the gap between communities: the need for developments in the interplay between climate scientists and spatial planners

The knowledge, the instrumentation and the expertise developed over the last decades regarding land surface-atmosphere interactions and their impacts on local-to-regional climate and air quality could deliver operational and useful outcomes for policy makers and land planners, and thus benefits for populations, activities and ecosystems. One action that can help bridge this gap is to introduce (or re-introduce) climate expertise into the spatial planning process. The climate issue has clearly become one of the main priorities of planning authorities throughout the world (e.g., Bulkeley, 2006; Wilson and Piper, 2006 ; Davoudi et al., 2009) in response to the widespread call for fighting global change in many fields and scales of policy. However, relatively few planning authorities directly call upon climate experts. This absence of climate expertise leads planners to ignore many levers of action at local and/or regional scales, some of them being sketched throughout this article.

Nowadays, more and more urban planning authorities develop in-house climate expertise, with sometimes interesting results. For example, efforts are being made in an increasing number of cities to reduce the urban heat island effect (Ren et al., 2011; Cordeau, 2014). These additional climate skills are nevertheless largely dedicated to urban areas and consequently face difficulties to consider the influence of surface-atmosphere interactions at broader spatial scales. They generally hardly consider as well the interplay between climate and air quality issues. There are, however, a few cases that can be sources of inspiration. For instance, for the Stuttgart Metropolitan Area, which is 3654 km$^2$ wide, the City of Stuttgart's Department of Urban Climatology produced a climatic atlas, based on a climatope approach to assess the influence of spatial units with similar microclimatic characteristics on atmospheric conditions (Baumüller et al., 2008). This initiative resulted in urban and spatial planning guidance, with the objective to improve the flow of fresh air from the agricultural and natural areas and thus to refresh, clean up and prevent tempoerature inversion above built surfaces. The development of local-to-regional actions taking advantage of multiple surface-to-atmosphere interactions can hardly be conceived without using regional meteorological or climate models, since the same land-use or land management direction can have very different and even inverse consequences, depending on the context (Marshall et al., 2004a, Schneider and Eugster, 2007, Lobell and Bonfils, 2007, DeAngelis et al., 2010). An example of successful collaborations between communities is the digital modelling platform built within the framework of the ACCLIMAT project (https://www.umr-cnrm.fr/ville.climat/spip.php?rubrique47).

This platform allows the numerical modelling of different processes of the city system and their interactions. The developed physical- and urban-based models are forced by socio-economic scenarios of urban development and local climatic scenarios. It is then possible to produce different city projections, from the present-day to the end of the century, under different future climates conditions,

and to estimate the impacts of these cities on urban climate or on building energy consumption.

Another difficulty to develop a collaborative action lies, among others, in the spatial gap between the respective scales of reference of climate scientists and spatial planners. Climate models have not yet sufficiently been tested at the intermediate spatial scales that are generally considered by planners in their practice. Regional climate models often work at resolutions lower than 15 km x 15 km, while urban

climate models work on meshes of about 1 km x 1 km. There is therefore a need to develop models functioning at intermediate scales and integrating a description of land surfaces closer to the definitions and representations used by spatial and urban planners.

Lastly, we need to give more attention today to the modifications created by land-use management (e.g. agricultural and forestry practices) on top of land-use at a regional and global scale. For climate

scientists, this means to identify levers of action, among those proposed by practitioners, in terms of land-use management that can influence climate and air quality. For planners, this is another challenge emerging, questioning the contours of their field of activity, the discipline focusing historically on land-use and surface occupancy.

**6. Conclusion**

Land-atmosphere interactions involve many physical, biological and chemical processes that can all influence each other, and that are driven by the characteristics of the environment in which they take place (meteorological conditions, surface properties, etc.). To properly investigate the role and impact of land-atmosphere interactions, especially in the context of LULCCs, on local-to-regional climate and air quality, the most appropriate and comprehensive tools are required. It is difficult today to design

experimental protocols at the regional scale that allow us to identify interactions and impacts of specific processes. When modelling such interactions, one has to recognize that the description of land-use and land-management (areas concerned, type of crops, quantity of fertilizers used and actual seasonality of application, etc.), including surface properties and emission sources, are overly simplified in today's models. Not taking into account the land-surface characteristics certainly biases our projections.

Moreover, land-atmosphere interactions are often specific to the target landscape, especially at a local/regional scale; therefore, in this perspective, one can hardly propose general solutions or recommendations. Hence, there is a crucial need for a consistent description of surface characteristics in numerical tools, to both improve our knowledge and provide more appropriate information to urban/land-planners and stakeholders at the territory/local scale. Urban and peri-urban areas are of

particular attention in this context since land transformation can have big environmental impacts and affect the health and life of million people, given the human density in these areas. For example, there

is space for considering the links between atmospheric chemistry and land-atmosphere interactions, as a decision parameter for land-management, helping to maintain air quality and supporting ecosystem functioning. This leads us to touch on the notion of Ecosystem Services, which is an integrated approach that allows to effectively analyse and examine the ecosystem conditions in terms of whether or not the desired services are being delivered. Ecosystem services are highly interlinked, and any kind of human influence on the functioning of one service will likely have a large number of knock-down effects on other services. The types of ecosystem services dealing with the climate and the atmosphere come under the category of regulating services, which were identified and categorized in several studies (Cooter et al. 2013, Thornes et al. 2010). Nevertheless, the feedbacks of the atmosphere to the ecosystem functioning potentially affect the ability of those ecosystems to provide services to human population.

**Author contribution**

Raia Silvia Massad and Juliette Lathiere equally contributed in the conception, outline design, writing and revision of the manuscript.

Susanna Strada contributed to writing parts of the manuscript relative to physical and chemical processes
and to revising the manuscript.

Nathalie de Noblet has solicited this review in the context of the LabEx BASC, and participated in the conception of the manuscript and contributed to writing the discussion.

Marc Stefanon and Patrick Stella contributed to writing parts of the manuscript relative to physical processes.

Sophie Szopa contributed to writing parts of the manuscript relative to chemical processes.

Erwan Personne contributed to writing parts of the manuscript relative to biological processes.

Mathieu Perrin contributed to writing parts of the manuscript relative to urban planning, LULCC.

All authors participated in the outline design and reviewed the manuscript.

**Acknowledgements**

This work was supported by a grant overseen by the French National Research Agency (ANR) as part of the "Investments d'Avenir" Programme (LabEx BASC; ANR-11-LABX-0034). The work of Marc Stefanon was supported by the French National Research Agency (ANR) as part of the project Forewer (ANR- 14-CE05-0028).

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

Ziska, L. R. and George, K.: World Resource Review Vol. 16 No.4 Rising Carbon dioxide and invasive Noxious plantss: potential threats and consequences.,

**Table 1: Typical values of snow free albedo (α - %), Bowen ratio (β - %) and roughness length ($z_0$- m) for various surface land cover.**

|  | Bare soils | Grasslands | Forests | Crops | Urban areas |
|---|---|---|---|---|---|
| $\alpha$ | 0.14-0.28 (Matthews et al., 2003) | 0.17-0.25 (Matthews et al., 2003; Markvart et Castañer, 2003) | 0.08-0.18 (Matthews et al., 2003; Markvart et Castañer, 2003) | 0.13-0.25 (Matthews et al., 2003; Song, 1999) | 0.09 – 0.27 (Taha, 1997; Brazel et al., 2000; Santamouris, 2013) |
| $\beta$ |  | 0.4 (Teuling et al., 2010) | 0.9-1.6 (Teuling et al., 2010) |  | 1.5 – 5 (Oke, 1982; Oberndorfer et al., 2007; Pearlmutter et al., 2009) |
| $z_0$ | 0.02-0.04 (Matthews et al., 2003; Wieringa, 1993) | 0.11 (Matthews et al., 2003; Wieringa, 1993) | 0.91-2.86 (Matthews et al., 2003; Wieringa, 1993) | 0.05-0.18 (Matthews et al., 2003; Wieringa, 1993) | 0.5 – 2 (Kato and Yamaguchi, 2005; Foken, 2008) |

**Table 2: Synthesis of direct and indirect effects of land use and land cover changes as seen from a physical, biological or chemical perspective as well as land management changes as seen from a physical, biological or chemical perspective**

| | Process Involved | Some Affected Variables / Fluxes | Direct Effect | Cascading Effect | Scale concerned |
|---|---|---|---|---|---|
| Land_use Intensification | Physical | Albedo – Roughness Length / sensible and latent heat fluxes | Change in atmospheric momentum, heat and water content | Temperature – wind circulation – precipitation and cloud cover through convection processes | Global, Regional, Local |
| | Biological | Photosynthesis rate; Stomatal functioning; soil functioning (mineralization) | Change in atmospheric concentrations of GHG, non GHG, aerosols; and water content | Temperature; convection; cloudiness | Global, Regional, Local |
| | Chemical | Ecosystem emission capacity, leaf area, deposition efficiency on surfaces | Change in net emission fluxes of chemical compounds to the atmosphere (VOCs) | Changes in atmospheric chemical composition ($O_3$, VOCs concentrations and reactions involved, aerosols) | Global, Regional, Local |
| Agricultural Intensification | Physical | sensible and latent heat fluxes – Albedo – Roughness Length | Change in atmospheric momentum, heat and water content | Temperature – wind circulation – precipitation and cloud cover through convection processes | Global, Regional, Local |
| | Biological | Change in Photosynthesis; change in productivity | Change in GHG emissions | Change in net emissions of chemical compounds ($NH_3$, BVOCs), change in water and energy budgets | Regional, Local |
| | Chemical | Change in N input | Change in emissions of N compounds from soils ($NO_x$ and $NH_3$ for example) | Impact on primary and secondary aerosol formation | Regional, Local |
| Urban Intensification | Physical | Sensible and latent heat fluxes – Albedo – Roughness Length | Change in atmospheric momentum, heat and water content | Temperature – wind circulation – precipitation and cloud cover through convection processes | Global, Regional, Local |
| | Biological | Change in Biodiversity | Change in photosynthesis and productivity | Change in plant phenology due to temperature and water availability | Regional, Local |
| | Chemical | Emissions sources of chemical compounds in the atmosphere (amount & composition), deposition efficiency on surfaces | Change in atmospheric chemical composition, occurrence of pollution episodes, increase of background pollution | Natural and agricultural ecosystem productivity affected by impacts on ecosystem functioning and stomatal closure | Regional, Local |

**Appendix**

In this appendix we present the physical, chemical and biological theoretical backgrounds for the different interactions presented in this paper.

## 1. Physical processes

The different types of surfaces covering the earth tightly control (micro-) climate through their influences on the radiative budget, the energy balance, the water balance, and air flows. The radiative budget determines the energy received by the surface. For any surface, the net radiation ($Q^*$) is defined as follows:

$$Q^* = (S \downarrow - S \uparrow) + (L \downarrow - L \uparrow) \tag{1}$$

where $S$ and L are short- and longwave radiations, respectively, and $\uparrow$ and $\downarrow$ refer to upwelling and incoming components, respectively. By considering surface albedo ($\alpha$) and surface and air emissivities ($\varepsilon_s$ and $\varepsilon_a$, respectively), surface temperature ($T_s$) and high altitude air temperature ($T_a$), Eq. (1) becomes:

$$Q^* = (S \downarrow - \alpha S \downarrow) + (\varepsilon_a \cdot \sigma \cdot T_a^4 - \varepsilon_s \cdot \sigma \cdot T_s^4) \tag{2}$$

with σ the Stefan-Boltzmann constant ($= 5.67 \times 10^{-8}\,\mathrm{W\,m^{-2}\,K^{-4}}$). The energy balance for any surface is linked with its radiation budget through $Q^*$ and can be expressed as (assuming there is no energy stored with land, which excludes therefore permafrost regions or regions with snowy winters, for example):

$$\Delta Q_S = Q^* - H - -G \tag{3}$$

where $\Delta Q_S$ is the change of energy within the considered surface layer, $H$ is the sensible heat flux (dry heat convectively exchanged between the surface and the atmosphere, that changes both the emitter and receptor temperatures), $LE$ is the latent heat flux (i.e., energy dissipated during evapotranspiration, water vapour convectively exchanged between the surface and the atmosphere, that changes both the emitter and receptor moisture conditions). LE includes both water evaporation ($E$) (i.e., from soil, dew, water interception by leaves, lakes and oceans) and plant transpiration ($T$). $G$ is the conductive ground heat flux from/to deeper layers. $G$ is often small and negligible for minor scales compared to $H$ and $LE$ fluxes. The energy and water balances are connected through the evapotranspiration (i.e., the sum of $E$ and $T$). The water balance for a surface including vegetation without considering lateral exchange between adjacent soil volumes can be expressed as:

$$\Delta S = P - E - T - R - D \tag{4}$$

where $\Delta S$ is the change of water content within the given layer, $P$ is the precipitation (in case of surface layer) or percolation from the above layer, $R$ is the surface runoff, and $D$ is the drainage. Note that the term $\Delta S$ includes soil moisture, surface water, snow, ice cover, and — depending on the depth of the considered soil layer —

groundwater. It also includes the interception storage. Finally, any convective fluxes between the atmosphere and the surface in the surface boundary layer can be expressed following the flux-profile relationships as:

$$F_\chi = -k \cdot u_* \cdot \frac{\chi_z - \chi_s}{ln\left(\frac{z-d}{z_0}\right) - \psi_\chi(z/L)} \tag{5}$$

where $F_\chi$ is the bi-directional land-atmosphere turbulent flux of the scalar $\chi$ (e.g., temperature, water vapor, carbon dioxide, ozone), $k$ is the von Karman constant (= 0.4), $u_*$ is the friction velocity, $z$ is the height above ground, $d$ is the displacement height, $z_0$ is the roughness length, and $\psi_\chi(z/L)$ is the stability correction function accounting for atmospheric stability.

From the previous equations, it can be seen that any LULCC induces modifications in the surface radiative, energy, and water budgets, which may in turn modify the climate. First, the energy received by the surface is closely related to surface properties (i.e., $\alpha$ and $\varepsilon_s$) (Eq.2). Any darkening (brightening) of the surface by LULCC will decrease (increase) albedo and make more (less) energy available at the surface. This alteration may result in increased (decreased) surface and air temperature. Similarly, any increase (decrease) in surface emissivity due to LULCC modifies the radiative budget of the surface resulting in the decrease (increase) in surface and air temperature. Typical values of albedo ($\alpha$), Bowen ration ($\beta$) and roughness length ($z_0$) are summarized in **Erreur ! Source du renvoi introuvable.**. Then, LULCCs modify the energy dissipation which occurs mainly through turbulent fluxes (*H* and *LE)* (Eq. 3), and the partitioning between H and LE that is often characterized by the so-called Bowen ratio (i.e., $\beta$ the ratio *H/LE*, see Table 1 for typical values). This latter varies with surface properties: the largest the amount of evapotranspirative surface is, the lowest the Bowen ratio is. The Bowen ratio is controlled by the presence/absence of free water (e.g., lakes, oceans, rivers, soils) and as well by the presence/absence of vegetation (e.g., surface, density, phenology) and its physiological activity particularly linked with stomatal conductance (for details about the factors affecting stomatal opening and closure, see part 2 below on biological processes). The partitioning of turbulent heat fluxes influences local climatic conditions, especially air temperature: while a large Bowen ratio (i.e., *H >> LE*) induces local warming of the ambient air with consequences and feedbacks on ecosystem functioning (e.g., thermal stress) and air pollution (e.g., chemical production/depletion in the atmosphere), a small Bowen ratio due to larger *LE* allows surface cooling as energy is converted into latent heat, followed by air cooling as H is reduced. Yet, it also influences the water balance due to its link with *LE* (Eq. 4). Finally, although convective fluxes are closely related to local climatic conditions (e.g., wind speed and temperature influencing $u_*$ and atmospheric stability respectively), surface largely influences the efficiency of convective fluxes through its impacts on $d$ and $z_0$ (Eq. 5). Increasing surface roughness (e.g., through afforestation) enhances turbulent exchanges owing to the increase in $d$ and $z_0$, and conversely. In a general manner, the higher the canopy is, the larger $d$ and $z_0$ are (Table 1), even if they are influenced by other parameters (e.g., LAI for pseudo-natural ecosystems, building density for urbanized areas). However, it must be kept in mind that land-atmosphere exchanges are also dependent on scalar concentration difference between the surface and the

atmosphere, meaning that air mass composition (e.g., temperature, $H_2O$, $CO_2$ or pollutants) and surface emissions (e.g., from manure application or anthropogenic activities) are crucial variables. Yet, plants can absorb or emit various compounds according to their development and functioning in link with meteorological conditions. In turn, the magnitude and direction (i.e., from or to the atmosphere) of the fluxes will affect the atmospheric composition.

2.  **Biogeochemical processes at the land/atmosphere interface**

Biological activity occurs in soils and within the vegetation. It affects number of physical, chemical and biogeochemical processes and therefore also the exchanges between land and atmosphere.

**Soil** microbial activity primarily involves the production of energy by autotrophs through aerobic respiration. Organisms can obtain energy through anaerobic respiration that originates from the reduction of organic compounds, such as fermentation, or inorganic compounds, such as nitrate reduction, denitrification or methanogenesis. The oxidation of certain minerals, also called chemolithotrophy, can also be a source of energy for living organisms such as nitrification, anammox (ammonium anaerobic oxidation) or sulphur oxidation. As all metabolic pathways, environmental factors such as temperature, water presence or absence, and substrate availability control those processes and are therefore affected by LULCC. The different metabolic pathways release into the environment different reactive gases ($NH_3$, $NO_x$, BVOCs) and non-reactive (or less reactive) organic or mineral compounds ($CO_2$, $N_2O$, $CH_4$, $H_2O$) affecting the atmospheric composition. These compounds can have chemical (see Sect. 3.3) or physical effects (see Sect 3.1 change in the water and energy budget) and/or warming effect. In turn those atmospheric changes feed back on ecosystem functioning through direct and indirect effects.

**Plants** are considered as heterotroph and can therefore convert sunlight and $CO_2$ into organic carbon through photosynthesis. One of the major actors in photosynthesis is the stomatal movement, which allows the leaf to change both the partial pressure of $CO_2$ at the sites of carboxylation and the rate of transpiration interlinking the water and carbon budgets. Another important actor of photosynthesis is RuBisCO, the major enzyme involved in the fixation of $CO_2$. RuBisCO is a rate-limiting factor for potential photosynthesis under the present atmospheric air conditions (Spreitzer & Salvucci 2002). It contains relatively large amounts of N, accounting for 10 to 30% of total leaf N-content for C3 type plants and 5-10% of total leaf N for C4 type plants it is thus an important link between the C and N cycles in vegetated surfaces (Makino, 2003; Carmo-Silva et al. 2015).

The plant's photosynthetic enzymes and the functioning of the stomata are affected by: (i) changes in the physical environment of the leaves (water potential, temperature, and $CO_2$ concentration; Farquhar and Sharkey, 1982); (ii) contact with atmospheric chemical pollutants (oxidative gaseous compounds, nitrogen deposition); (iii) availability of other resources (nitrogen, phosphorous); and (iv) interaction with adjacent living organisms (competition for resources, invasion by pests). Climate change or land use and land cover changes can directly or indirectly modify all these factors. Moreover, stomatal conductance plays a major role in the surface energy budget when plants are involved, as explained in Sect. 3.1, and can be one of the pathways of feedbacks between the

atmosphere and the surface since they regulate $CO_2$ input to the leaves and water output from the leaves. Vegetated surfaces are also involved in the exchange of other reactive species such as $NH_3$ in fertilized agricultural land as well as volatile organic compounds (VOCs) as a communication or defence tool that facilitate interactions with their environment, from attracting pollinators and seed dispersers to protecting themselves from pathogens, parasites and herbivores (Dudareva et al., 2013).

Some examples of how LULCC can affect climate through biological activity of soils and plants are given below:

- **Water Use Efficiency (WUE)** is defined as the ratio between the rates of carbon assimilation (photosynthesis) and transpiration. Plants that can have a lower transpiration rate without simultaneously decreasing their photosynthesis and thus biomass production are a desired trait in crop production. C4 type photosynthetic plants as opposed to C3 type photosynthetic plants have the capacity to concentrate $CO_2$ in their mesophyll cells and can therefore have a higher WUE. Plants in general respond to changing $CO_2$ concentrations, for example, it has been shown that an increased $CO_2$ concentration tends to reduce stomatal conductance while still increasing photosynthesis (Ainsworth and Rogers, 2007). This has several implications when considering different land uses in the context of climate change and increased $CO_2$ concentrations. Recent research demonstrate that most of C4 plants almost certainly display increasing water-use efficiency with increasing $CO_2$ concentrations, which allows them to better deal with conditions of water stress (Maroco et al., 1999; Conley et al., 2001). Consequently, this phenomenon should allow plants, in the future, to grow in areas where they currently cannot survive due to limited soil moisture availability. Those same plants will also be able to better resist drought periods and heat waves (Prior et al., 2011; Aparicio et al. 2015). WUE issues can be artificially overcome by irrigation, with consequences on plant phenology and local climate. Intensification of the water cycle or increased drought conditions because of climate change and LULCC modify the biological functioning of the soil-vegetation system and lastly influence the local climate.

- Increased **temperature** and frost-free days as well as atmospheric $CO_2$ concentrations affect the activity of RuBisCO. As a result, the growing season elongates and, if no other limiting factors are present, the net primary production (NPP) increases accordingly (Reyes-Fox et al. 2014; Fridley et al. 2016), which could be beneficial in temperate regions. However, longer growing seasons increase pressure on the water cycle therefore affecting local climate and resulting in potentially negative feedbacks on the carbon cycle (Wolf et al. 2016; Ciais et al. 2005). Due to temperature effects, species migrate to higher latitudes or altitudes (Hillyer and Silman, 2010; Brown et al. 2014; Spasojevic et al. 2013) resulting in LULCCs, changes in emissions of reactive trace gases and in habitat for biodiversity. Finally, higher temperatures enhance soil microorganism activity leading to higher mineralisation rates and consequently $CO_2$ release to the atmosphere.

- Rate of photosynthesis is directly correlated to leaf nitrogen content on a mass basis. **Nutrient Use Efficiency (NUE)** is defined as the ratio between the amount of fertilizer N removed from the field by the

crop and the amount of fertilizer N applied. Increased anthropogenic synthesis of mineral fertilizers to intensify crop production impairs the global N cycle as illustrated by the N-cascade (Galloway et al., 2003; Fowler et al., 2013) with impacts on biodiversity (Sutton et al., 2011), water and air quality (Billen et al., 2013; Erisman et al., 2013), and productivity and nutrient cycling (Phoenix et al., 2003; de Vries et al.,

2009). Nitrogen and carbon cycles are interlinked through biosphere-atmosphere interactions via biological processes, as detailed here, and chemical processes in the atmosphere, as detailed in Sect. 3.3. Nitrogen is a limiting nutrient for plant growth. In the tropics, warmer and wetter climate induces high soil mineralization and biological fixation (Cleveland and Townsend, 2006; Yang et al., 2010) therefore increasing N availability; however, this is not the case in mid- and high-latitude regions. Increasing N

availability to vegetated surfaces raise NPP, at least temporarily, with increased C storage in soils and higher N values in the vegetation (Yue et al. 2016) with direct effects on climate but also indirect effects via impacts on the water and energy budgets of certain areas.

-    Another example is the effect of elevated biotic or abiotic stress on plants. Increased **ozone** concentrations is a typical example, which affects stomatal conductance and photosynthesis (Fowler et al., 2009; Reich

and Lassoie, 1984). Ozone is a strong oxidant that can alter the functioning of plant cell in different ways. At relatively high concentrations, we observe: (i) direct damage of leaf epidermis cells (Sandermann et al., 1997; Günthardt-Goerg et al., 2000), (ii) modification of stomatal resistance via damage of guard cells causing leaky stomates (Paoletti and Grulke, 2010; Wittig et al., 2007), and (iii) alteration of cell walls and cell membranes (Gunthardtgoerg and Vollenweider, 2007). At low concentrations, we observe also

negative effects: (iv) ozone penetration to the mesophyll cells enhances production of reactive oxygen species (ROS) (Schraudner et al., 1998; Wohlgemuth et al., 2002), and it can also alter certain proteins and enzymes therefore affecting plant photosynthesis and biomass production (Heath, 1994). It is important to note that there is an accumulative effect of exposure to ozone concentrations by the plant (Fuhrer et al., 1997; Super et al., 2015). Different stresses affect different plant functioning but in most

cases they induce the production of ROS and the emissions of biogenic VOCs with consequences on air quality.

In summary, the major biologically driven interactions from a LULCC or LMC perspective between the atmosphere and the terrestrial biosphere result from the following changes. (i) The total productivity of the ecosystem as affected by changes in photosynthesis and soil microorganism activity and conditioned by the

availability of water and nutrients (N) thus resulting in the release or absorption of $CO_2$ to/from the atmosphere. (ii) Enhanced exchange of reactive trace gases ($NH_3$, BVOCs, $NO_x$) and their subsequent impact on nutrient availability in ecosystems and air quality. (iii) The indirect impacts of plant productivity on the energy and water budgets locally and regionally and their subsequent impacts on local and meso climates. In the sections below, we discuss some examples of these biological interactions as influenced by three LULCC and LMC.

### 3. Chemical processes in the atmosphere

Terrestrial ecosystems are both sources (nitrogen and organic species, particles) and sinks (ozone for instance through deposition on vegetative surfaces) of chemical compounds. Along their life, even trace amounts of these reactive gaseous and particulate matter (called aerosols) interact and influence the Earth system at large scales, regarding climate evolution, and at regional-local scales, regarding air quality. Air pollutants, both gases and aerosols, threaten human and ecosystem health and can be directly emitted (primary pollutants), or produced by reactions between primary pollutants (so called secondary pollutants). Any modification in the landscape structure, land-use or land management therefore has the potential to modify the air chemical composition. Some agricultural practices are shown in the literature to affect air quality. This is the case of fertilization as a source of ammonia, fires as a source of ozone precursors and aerosols, or fallow periods as a source of coarse aerosols. In this section we will focus especially on secondary pollutants such as ground-surface ozone ($O_3$) and secondary aerosols, that strongly affect air quality and whose production, lifetime and deposition involve the terrestrial biosphere, as demonstrated in several publications.

- **Compound emissions**

Natural sources contribute 90% of global annual VOC emissions (BVOCs, mainly from vegetation, with a minor contribution from oceans), while anthropogenic source (AVOCs, e.g., motor vehicle exhaust, solvents, biomass burning) only contribute 10% (Simpson et al., 1999). VOCs include thousands of different species. Among BVOCs, isoprene and monoterpenes are the most abundant, with isoprene that contributes around 50% of the total BVOC emissions and is mainly released by tropical and temperate vegetation, whereas monoterpenes contribute around 15% and are mostly emitted by boreal vegetation (Arneth et al., 2008). These secondary metabolites have been shown to play an important role for plants (thermotolerance, plant protection against abiotic stressors, plant-plant or plant-insect communication, etc.) (e.g., Peñuelas and Llusià, 2003). Broadleaf and needle-leaf forests are usually much stronger BVOC emitters compared to crops and grasslands. Temperature, radiation, water stress and atmospheric $CO_2$ concentration are strong external drivers of BVOC emissions (Peñuelas and Staudt, 2010). With a lifetime of a few minutes to hours, BVOCs are very reactive gases that play an important role in photochemistry (i.e., $O_3$ production), and contribute to the formation of biogenic secondary organic aerosols (BSOAs) (Atkinson and Arey, 2003).

Agricultural fertilization and natural soil processes of nitrification and denitrification are a significant source of nitrogen compounds, such as nitrogen oxide (NO) and nitrogen dioxide ($NO_2$). These two compounds are treated as a unique family (i.e., nitrogen oxides, NOx) due to the rapid cycling between NO and $NO_2$ during daytime (about one minute), while the NOx family is mainly composed by $NO_2$ at night-time. Overall, the lifetime of NOx is approximately one day. At the global scale, NOx are mainly emitted by anthropogenic sources (e.g., fossil fuel combustion, biomass burning) and more moderately by lightning.

- **Surface ozone**

Ozone is a highly reactive compound that is present in the stratosphere, where it protects life on Earth from ultra-violet (UV) radiations, in the troposphere and close to the surface, where it threatens human and plant health due to its oxidizing effect on living tissues. Ground-surface $O_3$ has a lifetime of one month and is mainly formed on sunny and warm days because of a complex and non-linear interplay between NOx and VOCs (Sillman, 1999). Surface $O_3$ production relies on the imbalance between $O_3$ production via $NO_2$ photolysis (i.e., NO reactions with peroxy radicals, $HO_2$) and $O_3$ removal via reaction of $O_3$ with NO. Organic peroxy radicals (i.e., $RO_2$) from the oxidation of VOCs in forested (BVOC-dominated) or highly polluted (AVOC-dominated) regions also contribute to $O_3$ production. While $O_3$ removal depends on $O_3$ photolysis, reactions with radicals (e.g., OH and $HO_2$) in remote regions, and dry deposition. The $O_3$ chemistry is characterized by two different photochemical regimes, driven by NOx and VOC concentrations: the NOx-sensitive regime, with relatively low NOx and high VOC concentrations, where $O_3$ increases with increasing NOx levels, with low sensitivity to VOCs; the VOC-sensitive regime, where $O_3$ increases with increasing VOC levels and decreases with increasing NOx (Sillman, 1999). Natural and anthropogenic ecosystems can therefore both influence the level of ozone concentration in the atmosphere, as sources of compounds involved in the ozone cycle, and be impacted by the ozone oxidizing effect, depending on the pollution level.

- **Secondary aerosols**

Atmospheric aerosol particles originate from a large variety of natural and anthropogenic sources. While primary aerosols are directly emitted as liquid droplets or solid particles (e.g., mineral dust, sea salt, pollen, black carbon from diesel engines or biomass burning), secondary aerosols result from gas-to-particle conversion. Secondary aerosols include inorganic (e.g., sulfate, nitrate) and organic species (named organic aerosols, OA), each species typically contributing about 10–30% of the overall mass load. However, both location and meteorological conditions strongly influence the air composition and the relative abundance of different aerosol types (Tunved et al., 2005; Deng et al., 2012).

In the last two decades, BVOCs have been identified as precursors of BSOAs, with monoterpenes and sesquiterpenes having a large potential to produce BSOAs (Kanakidou et al., 2005). Isoprene has a minor aerosol production yield but still significantly contributes to BSOA mass due to its abundance over total BVOC emissions and its large global source, especially during summer (Carlton et al., 2009). BSOA production shows a high variability that depends on external factors such as temperature and relative humidity (both playing a minor role), organic aerosol loading (which controls gas-particle partitioning of semi-volatiles), oxidants (which controls the extent and rate of reactions) and NOx levels. Carlton et al. (2009) observed the lowest SOA yields under "high NOx" conditions, whereas "NOx-free" conditions led to the highest measured SOA yields. Being involved into the absorption and scattering of radiation (direct effect) and into the alteration of cloud properties (indirect effect), BSOA, and SOA in general, can influence the radiative balance of the Earth, and therefore influence climate

(Forster et al., 2007). However, the exact contribution of BSOA to the radiative forcing is still very uncertain (Scott et al., 2014).

To form secondary aerosols, gas-to-particle conversion begins in the atmosphere with the oxidation, usually sustained by sunlight, of high volatility precursor gases (e.g., $SO_2$, NOx and VOCs, emitted especially from terrestrial ecosystems) into low volatility gases (e.g., sulfuric and nitric acid, ammonia, organics) that nucleate into stable molecular clusters (the ultra-fine mode, $10^{-3}$–$10^{-2}$ mm size range). Depending on ambient conditions, aerosols can still grow in size via condensation of gases onto the nucleated aerosol or coagulation (i.e., collision of two aerosols). The final aerosol size strongly determines multiple aerosol properties such as the interaction with radiation, impacts on human health, and aerosol lifetime and sinks. Typically, secondary aerosols belonging to the fine-mode have an atmospheric lifetime of about one-two weeks and can be removed from the atmosphere mainly via wet deposition (also termed scavenging), while coarse-mode aerosols, such as primary aerosols, are efficiently removed by dry deposition.

Among secondary aerosols, sulfates, nitrates and ammonium are produced primarily from atmospheric chemical reactions involving, respectively, sulfur dioxide ($SO_2$, mainly emitted from fossil fuel and biomass burning), NOx and ammonia ($NH_3$, largely emitted by domestic animals, synthetic fertilizers, biomass burning, and crops). Over half of atmospheric $SO_2$ is converted into sulfates, and half of emitted $NH_3$ is converted into ammonium aerosols. Together with nitrates, ammonium represents the main form of atmospheric nitrogen aerosols and may provide nutrients to vegetation growth in nitrogen limited systems (Mahowald et al, 2011). It is also worth mentioning phosphorus, a nutrient that plays a key role for many living organisms and is mainly present in the atmosphere in the aerosol mode. However, among atmospheric aerosols, the phosphorus composition, together with its size, geographical distribution and emission sources remain poorly characterized and investigated (Furutani et al., 2010).

Organic aerosols altogether contribute ~20–50% of the total fine aerosol mass at mid-latitudes and 90% in tropical forested regions (Kanakidou et al., 2005). Depending on the season and the location, secondary organic aerosols (SOAs) contribute 20–80% of measured mass of OAs.