# Peer review of "Reviews and syntheses: Influences of landscape structure and land uses on local to regional climate and air quality"

_Biogeosciences, 2018_

## Referee Comment (RC1) · Anonymous Referee #1 · 3 Dec 2018

Review of bg-2018-419: Reviews and syntheses: influences of landscape structure and land uses on local to regional climate and air quality

Summary The authors aim to review how land use affects climate and air quality. The present a framework for categorizing land use, and apply it to different aspects of the earth system, in order to demonstrate relationships between biosphere, climate, and air quality. They also present current challenges to understanding the effects of land use on air quality, provide suggestions for different groups to work together to address these challenges.

Overall impression

[Figure]

I appreciate the effort to synthesize this information, and I generally agree with the abstract, but this paper falls very short of its goals. It is poorly organized, repetitive, inconsistent in its application of the framework, contains enough awkward language up front that the goals of the paper are not clear, and the conclusions do not follow from the information presented. Additionally, there isn't a unifying story to make sense of the extremely diverse information presented. For example, the air quality aspect is in the title and highlighted in the abstract, but is not addressed until page 23, and then only elaborated upon in a few sections. Also, main conclusions focus on model shortcomings, but there isn't a modelling review included.

It seems to me that this is about 5 papers mashed into one. Most of the text reads like a list, and links and meaning across the different sections and information are not made. The middle section alternates between pedantic textbook material and an uncritical presentation of many studies. I suggest that the authors think about what point they want to make, and focus on that point. For example, the material is here for a review of land use and air pollution. But there is a lot extra physics information that doesn't need to be presented in order make the point that the physical processes are an influence. Here are some suggestions:

1) The introduction and framework are general and vague. Make a solid, but concise, assessment of land use/cover change as a foundation, with literature to back it up, and move on to the effects you want to review.

2) The land categorization is applied inconsistently, probably due to its overlapping nature. Maybe delineate by urban, agriculture, and other lands. Part of the confusion and repetition arises because the urban changes are land cover changes, but these two have been separated.

3) The physical, biological, chemical distinction does not work. The physical isn't related to the rest (except in 4.3 where is it relevant and sufficient), and the biological and chemical are both about biological emissions, with a fuzzy distinction between primary

biogeochemistry (co2, ch4, n2o), and trace gases and aerosols. Stick to the chemical species you are interested in, and organize them around the land categories or land changes.

4) Develop meaning and relationships through the presentation of the literature. Having separate discussions later, or pedantic explanations before, leaves the information essentially as a list, and the later discussions become repetitive and do not have the references to back up statements.

5) To show integration of processes, put them in the same section. Present evidence for each one and the evidence for how they interact. Segregating them by section makes it difficult to make linkages without repetition. For example, section 4.3 starts to tie together vegetation, boundary layer, and air pollution, and is understandable without the lengthy textbook sections in part 3.

6) To criticize models, you need a model review.

7) To make conclusions about what is lacking, the gaps and limitations of existing work need to be explained in the review, rather than listing all the literature results as facts. Currently, the paper reads like everything has been figured out, but the conclusions state that hardly anything has been figured out. The shortcomings of the models are not reviewed, but are presented as main conclusions.

Specific comments/suggestions:

Abstract (page 1)

physical, chemical, and biogeochemical land-atmosphere interactions is a very broad topic, while the paper focuses on air quality impacts of land use. the abstract needs to be clear about the focus of the paper.

line 18: "at" to "and"

line 27: "non-existent" is more common than "inexistent"

line 32: delete "but"

Introduction

The focus on urban/peri-urban and air quality is not clear. until the last few sentences This leaves the reader wondering why the rapidly growing body of literature on the effects of LULCC on the earth system is not accounted for (only a few papers are cited in the intro).

page 2, line 7: Not sure that anthropized is a word. Anthropogenic seems correct, although not usually applied this way.

page 2, line 9: reference for energy balance?

section 2 and use and intensification - this does not seem to be the appropriate title for section 2 - section 2 covers a lot more than this

page 4 lines 21-23: the section should start with this. it is unclear why the land use section starts with land cover. see comment above.

section 2.2: not much definition here. in fact you acknowledge that definitions vary considerably

page 5, lines 23-26: confusing- i am not sure what these numbers refer to.

page 6, lines 7-8: reference?

section 3.1 physical processes:

not sure all these equations are necessary. this whole section seems like a textbook. who is the audience? a shorter description of how things change is more meaningful. The description can cite various studies on these effects, and be more digestible by the reader. there are no citations in this section. actually, this section can be deleted because the next 3 sections are the ones that makes the point.

page 9, ine 12: incomplete sentence

section 3.1.1 land cover change

this section makes the case of the previous section.

section 3.1.2 agricultural intensification

page 12, line 29: and burning, and understory treatment, and different types of harvest, and planting

again, this section makes the point of section 3.1

section 3.1.3 urban intensification

this should probably be grouped with LULCC and it doesn't need all the references to the equations

section 3.2 biological

this textbook section is unnecessary as the following sections make the case

3.3 chemical

another long textbook section

3.3.1 land use intensification (page 23)

here is a review relevant to land change and air quality. But it reads more like a list than a review of evidence for making a point.

3.3.2 ag intensification

it seems like the previous ag section was cut short to put the information here

3.3.3 urban

page 23: what are the suggested trees in CA?

4 interactions

4.1 local to meso and 4.2 ecosystem

these don't seem to have any relation to air quality

4.3 air quality

this is relevant, and expands upon sections 3.3.1-3.3.3 but now you have mashed together the land type framework

5 discussion

5.1

this is a lengthy repetition of the previous reviews without the citations

5.2 challenges

page 34

you really haven't shown the linkages and feedbacks. the information is all compart-mentalized.

while your challenges are legitimate, it isn't clear how you reached them based on the previous reviews. there are little to no citations in this section, especially regarding modelling. you also don't acknowledge that models are essentially simplifications, and that they cannot contain every single detail, and that not every single detail matters for the desired outcome of a model.

the reviews need to make a critical assessment of the information, establishing meaning and caveats to the studies in this context, and showing where the gaps are, rather than listing all the information as fact. then you can use this analysis to develop your challenges.

5.3 interdisciplinary approaches

this is repetitive

6 bridging the gap

while this is important, i don't don't see how it relates to this article

7 conclusion

there isn't a modelling review, so i don't know how presenting all of the field research generates the conclusion that modelling is poor

---

## Referee Comment (RC2) · Anonymous Referee #2 · 4 Jan 2019

Assessment of paper "Reviews and synthesis: Influences of landscape structure and land uses on local to regional climate and air quality". By Massad et al.

Thank you for inviting me to review this paper. First and foremost, I would like to state that it is one of the most comprehensive manuscripts I have seen for some time. It has the potential to be a significant "go to" paper for anyone with interest in measuring or modelling land-atmosphere interactions. The reference list is – as might be expected for a review – very thorough, and it certainly alerted me to papers I was not aware of.

I think it is important first to note what the paper does not include. This is not to detract from it in any way at all, but simply to gain understanding as to what its main messages

are. The authors may like to consider a set of words to capture this, possibly towards the end of the Introduction. So not addressed in detail are:

(1) Many existing studies focus on the role of the land surface in mitigating carbon dioxide emissions. Significant effort is placed in closing the global carbon cycle, and there is a view that the land surface (and associated land-atmosphere CO2 exchanges under a changing climate) is where much uncertainty remains. In the most general terms, approximately 25% of CO2 emissions are believed to be drawn-down by the terrestrial ecosystems. Of concern is that this fraction may decrease into the future, especially through higher respirations or nutrient limitation. There is a small reference to this, indirectly, in Table 2 "Change in atmospheric concentrations of GHG". The title is clear, with no word "global" used, but it does mention "climate".

(2) Related to (1) above, much is described in the IPCC reports, and especially the recent 1.5°C and 2.0°C threshold assessment, about the role of BioEnergy with Carbon, Capture and Storage (BECCS). This form of large-scale geo-engineering of the global carbon cycle is not included here (for instance, the CO2 cycle is omitted from Schematic Figure 5).

(3) The paper is very much a qualitative assessment, with most display items more schematic in format. Hence, it is presently difficult to compare effects, and so the logical conclusion is the one that the authors present. That is, there is a need for an overall integrated tool that would allow effective intercomparison of regional effects, drivers and feedbacks.

(4) The major part of the paper concerns geochemical feedbacks, rather than the more physical one. There are some exceptions. For instance, one page 14, there are citations to papers describing how different land cover types have the potential to either suppress or aggravate any future extremes in a changed climate.

By stating something along the lines (1) – (4) will then make the paper stronger, as clearer then what the paper does encompass. Moreover, this is where I believe the

manuscript is very powerful indeed. It is arguably that of the range of environmental concerns, climate change has taken too much of the attention. Many of the more local/regional effects will be just as important to the individuals concerned. This is especially true of air quality, or strong local pollutants that threaten food security – both of which are either modulated by the land surface or impact on it.

This paper, for the first time, places emphasis on non-global pollutants, and it is revealing from much of the literature cited that the implications are likely to be large in many instances. Those who build the air quality and atmospheric tracer components of weather forecasts, regional climate models or even full Earth System Models will appreciate this manuscript, bringing the latest understanding of the terrestrial role into a single document.

This review is slightly different to usual because most papers have quantitative plots which can be assessed and studied in detailed, and then commented on. So really it is only possible to give an overview here. The authors can if they like, consider the points above and associated context-placing. As always with manuscripts, please read through carefully again – especially as now a break since submitting. The paper is very long, and so possibly look for any places where the writing can be tightened. As environmental science is evolving fast, it might be worth a quick, targeted literature search of any very recent 2018 papers on Scopus or the Web-of-Science. Otherwise, I think the document could be published almost in its current form.

A very small thing - the legends in Figure 2 are in small font – please make them slightly bigger.

---

## Author Comment (AC1) · 8 Feb 2019

We thank Referee #1 for their careful reading and useful comments. Both reviews have very conflictual views of our paper. Reviewer 2 thinks we have done a relatively good job and encourages us with a limited amount of suggested changes. Reviewer 1 rejects our work with a very critical point of view. A 3rd review would have been quite helpful in that sense.

Reviewer general comments

Summary: The authors aim to review how land use affects climate and air quality. They

present a framework for categorizing land use, and apply it to different aspects of the earth system, in order to demonstrate relationships between biosphere, climate, and air quality. They also present current challenges to understanding the effects of land use on air quality, provide suggestions for different groups to work together to address these challenges.

I appreciate the effort to synthesize this information, and I generally agree with the abstract, but this paper falls very short of its goals. It is poorly organized, repetitive, inconsistent in its application of the framework, contains enough awkward language up front that the goals of the paper are not clear, and the conclusions do not follow from the information presented. Additionally, there isn't a unifying story to make sense of the extremely diverse information presented. For example, the air quality aspect is in the title and highlighted in the abstract, but is not addressed until page 23, and then only elaborated upon in a few sections. Also, main conclusions focus on model shortcomings, but there isn't a modelling review included.

It seems to me that this is about 5 papers mashed into one. Most of the text reads like a list, and links and meaning across the different sections and information are not made. The middle section alternates between pedantic textbook material and an uncritical presentation of many studies. I suggest that the authors think about what point they want to make, and focus on that point. For example, the material is here for a review of land use and air pollution. But there is a lot extra physics information that doesn't need to be presented in order make the point that the physical processes are an influence

*Reply:

We will re-organize the manuscript to respond to the reviewers concerns according to the following:

- We will review the abstract so as to better reflect the objectives and the content of the manuscript

- We will complete the introduction with supplementary literature related to land-use, land cover changes, and update if necessary. In the Introduction, we will precise that this review is intended to be accessible to the specialists (i.e., mainly scientists) and non-specialists (e.g., land-planners, stakeholders and decision-makers).

- We will move the "textbook material" highlighted in section 3 to an appendix to provide fundamentals of physics, chemistry and biology for those who may not be familiar with the theory that is behind the reviewed studies.

- We propose to profoundly re-organize section 3. Rather than the "physical", chemical" and "biologeochemical" subsections further subdivided (repetitively) by land categories, we propose two main sub-sections organized by land use and land cover changes: Land cover changes (deforestation, wetland conversion, urbanisation), Land intensification (agricultural intensification and urban intensification)

- Within each section, we will separate the literature review according to "experimental" and "modelling" studies, and we will critically summarize the shortcomings of each. We will use the conclusions (Section 5.1: synthesis of current knowledge) to back up the different sections. We would like to express here that this is a substantial revision of our paper.

**Comment:The introduction and framework are general and vague. Make a solid, but concise, assessment of land use/cover change as a foundation, with literature to back it up, and move on to the effects you want to review.**

*Reply:The assessment of land use/cover change is already in section 2 of the manuscript (rather than in the introduction). We will add a section detailing the potential impacts of those changes on the atmospheric compartment (which will be a very brief summary of sections 3.1.1, 3.2.1 and 3.3.1, the details of which will be moved to an appendix section). For us, the introduction is here to present the specificities of our review compared to other existing reviews in the literature and to state the objectives of this review.

**Comment:The land categorization is applied inconsistently, probably due to its over-lapping nature. Maybe delineate by urban, agriculture, and other lands. Part of the confusion and repetition arises because the urban changes are land cover changes, but these two have been separated.**

*Reply: The difficulty is in the fact that we are not looking at land use but at land use changes. We will however make the effort to limit the categories we look at regarding the Urban changes. We will separate urban intensification (considered as a Land use change) from urbanisation (land cover change).

**Comment:The physical, biological, chemical distinction does not work. The physical isn't related to the rest (except in 4.3 where is it relevant and sufficient), and the biological and chemical are both about biological emissions, with a fuzzy distinction between primary biogeochemistry (co2, ch4, n2o), and trace gases and aerosols. Stick to the chemical species you are interested in, and organize them around the land categories or land changes.**

*Reply:We agree with your analysis. We will remove this distinction in our text and re-organise the paper by categories of land-use changes, detailing for each category the impacts on the atmosphere and differentiating "experimental results" and "modelling results" so as to include a review of models. Thanks to this re-organisation, we will show the links between different studies and we will be more critical of the results based on other results. However one objective of our paper is to clearly put forward the various ways land and atmosphere interact (via changes in physical processes, biological and chemical processes), and which are almost never considered together in climate models (nor global nor regional). This will be clarified in the discussion.

**Comment:Develop meaning and relationships through the presentation of the literature. Having separate discussions later, or pedantic explanations before, leaves the information essentially as a list, and the later discussions become repetitive and do not have the references to back up statements.**
*Reply: By re-organising the manuscript as explained in responses 1 and 3 and by moving the "textbook" material to an appendix, we think that the literature review will improve, will become less pedantic and more discursive, rather than a list. The section "synthesis of current knowledge" (5.1) will be merged within section 3 to follow on directly and synthesize the review done. This way the discussion section will come earlier in the manuscript and will be less repetitive.

**Comment:To show integration of processes, put them in the same section. Present evidence for each one and the evidence for how they interact. Segregating them by section makes it difficult to make linkages without repetition. For example, section 4.3 starts to tie together vegetation, boundary layer, and air pollution, and is understandable without the lengthy textbook sections in part 3.**

*Reply: We agree with the reviewer and, as stated above (Response 4), we will merge sections on physical, chemical and biological processes involved in land use changes and we will discuss their impacts in a more comprehensive way that looks at the interactive system as a whole.

**Comment:To criticize models, you need a model review.**

*Reply: A model review (or references to already existing model reviews) will be integrated in section 3 within each land cover change section, as it can be seen in the new table of contents below. Our intention is not to criticize models but rather to show that there are today two 'niches' that have not been sufficiently looked at: 1) the combined physical-biological-chemical effects of land changes on climate at all spatial scales, 2) the specific 'territorial' scale that is smaller than the continental one and larger than a single city.

**Comment:To make conclusions about what is lacking, the gaps and limitations of existing work need to be explained in the review, rather than listing all the literature results as facts. Currently, the paper reads like everything has been figured out, but the conclusions state that hardly anything has been figured out. The shortcomings of**

the models are not reviewed, but are presented as main conclusions.

*Reply: By re-organising section 3 and making a more critical review of the literature, we will be able to present more clearly the gaps and limitations of models as well as experimental results, and to move on directly to the discussion and conclusion sections. Again our main argument is not to say that everything has been figured out nor that 'hardly anything has been figured out' but that the main pieces of the puzzle have not yet been assembled, nor applied at the specific spatial scale we're targeting.

Specific comments/suggestions:

**Comment: Abstract (page 1) physical, chemical, and biogeochemical land-atmosphere interactions is a very broad topic, while the paper focuses on air quality impacts of land use. the abstract needs to be clear about the focus of the paper.**

*Reply: The abstract will be rewritten in order to make the focus of the paper clearer. We are surprised though that the reviewer gets the impression we are focusing on air quality while discussion on this only occurs in 1/3rd of the paper.

**Comment:The focus on urban/peri-urban and air quality is not clear. until the last few sentences This leaves the reader wondering why the rapidly growing body of literature on the effects of LULCC on the earth system is not accounted for (only a few papers are cited in the intro).**

*Reply: We will state at the beginning of the introduction the specificities of the manuscript to justify the choice of literature cited. As discussed above, there is no specific focus on air quality. Air quality is one of the 3 aspects we're targeting. Between our 2 reviewers, one has understood our point, not the other. We will work towards clarifying this in the updated version.

**Comment:page 2, line 7: Not sure that anthropized is a word. Anthropogenic seems correct, although not usually applied this way.**

*Reply: We do not believe "anthropogenic" is the correct term to be used. We will

replace anthropized by man-shaped.

**Comment:page 2, line 9: reference for energy balance?**

*Reply: A reference will be added

**Comment:section 2 and use and intensification - this does not seem to be the appropriate title for section 2 - section 2 covers a lot more than this**

*Reply: We will change the title to: Land Cover and Land Use changes: history, dynamics and challenges. We will also add text to this section explaining what is our understanding of agricultural intensification and urban intensification.

**Comment:page 4 lines 21-23: the section should start with this. it is unclear why the land use section starts with land cover. see comment above.**

*Reply: We agree and we will start this section with lines 21-23.

**Comment:section 2.2: not much definition here. in fact you acknowledge that definitions vary considerably**

*Reply: Title changed

**Comment:page 5, lines 23-26: confusing- i am not sure what these numbers refer to.**
*Reply: These numbers will better explained.

**Comment:page 6, lines 7-8: reference?**

*Reply: Reference will be added.

**Comment:not sure all these equations are necessary. this whole section seems like a textbook. who is the audience? a shorter description of how things change is more meaningful. The description can cite various studies on these effects, and be more digestible by the reader. there are no citations in this section. actually, this section can be deleted because the next 3 sections are the ones that makes the point.**

*Reply: This section will be moved to an appendix. Moreover, in the Introduction, we

will precise the reader to whom the paper is addressed to.

**Comment:this section makes the case of the previous section.**

*Reply: This section will be moved to an appendix

**Comment:page 12, line 29: and burning, and understory treatment, and different types of harvest, and planting**

*Reply: Sentence will be added.

**Comment:again, this section makes the point of section 3.1**

*Reply:This will be addressed by changing the organisation.

**Comment:section 3.1.3 urban intensification this should probably be grouped with LULCC and it doesn't need all the references to the equations**

*Reply: References to equations will be removed. The sections will be modified as stated above in the general replies.

**Comment:section 3.2 biological this textbook section is unnecessary as the following sections make the case**

*Reply: Agree and will be removed to an appendix.

**Comment:3.3 chemical another long textbook section**

*Reply: Agree and will be removed to an appendix.

**Comment:3.3.1 land use intensification (page 23) here is a review relevant to land change and air quality. But it reads more like a list than a review of evidence for making a point.**

*Reply: Title will be changed to land cover change. Content as stated above will be merged with other sections and will be synthesized to read less as a list.

**Comment:3.3.2 ag intensification it seems like the previous ag section was cut short**

to put the information here

*Reply: These sections will be merged as stated above.

**Comment:3.3.3 urban page 23: what are the suggested trees in CA?**

*Reply:We will specify which tree species are concerned.

**Comment:4.1 local to meso and 4.2 ecosystem these don't seem to have any relation to air quality**

*Reply: In this paper we focus on the effects of LULCC and LI on air quality but also on local climate. This will be clarified in the introduction.

**Comment:4.3 air quality this is relevant, and expands upon sections 3.3.1-3.3.3 but now you have mashed together the land type framework**

*Reply: We agree with you, however in this section we have chosen to look at the inter-actions from a landscape/regional perspective and to discuss how differently LULUCC and LI interact together within a spatial framework.

**Comment:5.1 this is a lengthy repetition of the previous reviews without the citations**

*Reply: This section will be removed and merged within the new section 3 as stated above.

**Comment:page 34 you really haven't shown the linkages and feedbacks. the informa-tion is all compart-mentalized. While your challenges are legitimate, it isn't clear how you reached them based on the previous reviews. there are little to no citations in this section, especially regarding modelling. you also don't acknowledge that models are essentially simplifications, and that they cannot contain every single detail, and that not every single detail matters for the desired outcome of a model.**

*Reply: By changing and better discussing the literature in the modified section 3 we believe this question will be addressed. We will acknowledge model specificities as

recommended.

**Comment:5.3 interdisciplinary approaches this is repetitive**

*Reply: We understand that the reviewer refers to the introduction of section 5.3. We will review this part and remove repetitive ideas.

**Comment:6 bridging the gap while this is important, i don't don't see how it relates to this article**

*Reply: In this section, we treat existing gaps between scientific communities and spatial planners, which is justified, in our view since we need models that are more robust and a more exhaustive representation of scenarios that account for the social and economic drivers in the system. On the other hand, we agree that the part focusing on policy makers is not in the focus of this review and will be removed. A phrase will be added relative to this in the conclusion.

**Comment:7 conclusion there isn't a modelling review, so i don't know how presenting all of the field research generates the conclusion that modelling is poor**

*Reply: We will rephrase the conclusion and refer to the modelling review sections added in the new version of the manuscript.

!!!!!!!!!!!!!!!!!!!!!!!!!!!!!!!!!!!!!!!!!!!!!!!!!!!!!!!!!!!!!!!!!!!!!!!!!!!!!!!!!!!!!!!!!!!!!!!!!!!!!!!!!!!!!!!!!!!!!!!!!!!!!!!!!!!!!!!!!!

Modified structure of our manuscript

1. Introduction

2. Land Cover and Land Use changes: history, dynamics and challenges

2.1. Historical perspective

2.2. Land Use and Land Cover Change

2.3. Land-use Intensification
3. Human driven land use and land management changes and their impact on climate and air quality

3.1. Land Cover Change

3.1.1 Experimental studies

3.1.2. Modeling studies

3.1.3. Critical synthesis of studies

3.2. Land intensification (agricultural and urban)

3.2.1 Experimental studies

3.2.2. Modeling studies

3.2.3. Critical synthesis of studies

4. Interactions between different land cover, uses and management over a mosaic landscape: impacts on land-surface exchanges

4.1. Local- to Meso-climate perspective

4.2. Ecosystem functioning perspective

4.3. Air quality perspective

5. Discussion

5.1. Challenges ahead

5.2. Towards interdisciplinary approaches: some examples

5.2.1. Urban – agricultural – natural triptych in a N pollution context

5.2.2. Urban greening – UHI - and impact on VOC / NOx / O3 loop

6. Bridge the gap between communities: the need for developments in the interplay

between climate scientists and spatial planners

6.1. Introducing/reintroducing climate expertise into the spatial planning process

6.2. More consideration for land-use management

7. Conclusion

---

## Author Comment (AC2) · 8 Feb 2019

We thank Referee #2 for their careful reading and useful comments. Both reviews have very conflictual views of our paper. Reviewer 2 thinks we have done a relatively good job and encourages us with a limited amount of suggested changes. Reviewer 1 rejects our work with a very critical point of view. A 3rd review would have been quite helpful in that sense.

**General Comments**

Thank you for inviting me to review this paper. First and foremost, I would like to state

that it is one of the most comprehensive manuscripts I have seen for some time. It has the potential to be a significant "go to" paper for anyone with interest in measuring or modelling land-atmosphere interactions. The reference list is – as might be expected for a review – very thorough, and it certainly alerted me to papers I was not aware of.

I think it is important first to note what the paper does not include. This is not to detract from it in any way at all, but simply to gain understanding as to what its main messages are. The authors may like to consider a set of words to capture this, possibly towards the end of the Introduction. So not addressed in detail are:

**Comment: Many existing studies focus on the role of the land surface in mitigating carbon dioxide emissions. Significant effort is placed in closing the global carbon cycle, and there is a view that the land surface (and associated land-atmosphere $CO_2$ exchanges under a changing climate) is where much uncertainty remains. In the most general terms, approximately 25% of $CO_2$ emissions are believed to be drawn-down by the terrestrial ecosystems. Of concern is that this fraction may decrease into the future, especially through higher respirations or nutrient limitation. There is a small reference to this, indirectly, in Table 2 "Change in atmospheric concentrations of GHG". The title is clear, with no word "global" used, but it does mention "climate".**

*Reply: In this review we choose to treat the effects on air quality and local climate. The impact on the global climate (GHG emissions) is not the focus of our study. We will add an explicit phrase in the introduction to better explain this point.

**Comment: Related to (1) above, much is described in the IPCC reports, and especially the recent 1.5°C and 2.0°C threshold assessment, about the role of BioEnergy with Carbon, Capture and Storage (BECCS). This form of large-scale geo-engineering of the global carbon cycle is not included here (for instance, the $CO_2$ cycle is omitted from Schematic Figure 5)**

*Reply: Again as for global climate, the C cycle is not the focus of our study. This topic has been already extensively reviewed, even though we agree that there are still a lot

of gaps and challenges ahead. We will specify this point in the Introduction and will refer the interested readers to recent reviews (e.g., Le Quere et al., 2018; Saunois et al., 2018).

**Comment: The paper is very much a qualitative assessment, with most display items more schematic in format. Hence, it is presently difficult to compare effects, and so the logical conclusion is the one that the authors present. That is, there is a need for an overall integrated tool that would allow effective intercomparison of regional effects, drivers and feedbacks.**

*Reply: We agree with this analysis, and, upon suggestion from reviewer R1, we have re-organised the manuscript so as to better show how those conclusions are reached.

**Comment: The major part of the paper concerns geochemical feedbacks, rather than the more physical one. There are some exceptions. For instance, one page 14, there are citations to papers describing how different land cover types have the potential to either suppress or aggravate any future extremes in a changed climate.**

*Reply: These citations are presented in the section talking about physical effects. This section will be merged with others according to Land Cover Changes (LCC) and Land Intensification (LI). We will in this sense try to homogenize with other effects and consider if this is a point that should be more thoroughly reviewed or removed from our manuscript

**Comment: By stating something along the lines (1) – (4) will then make the paper stronger, as clearer then what the paper does encompass. Moreover, this is where I believe the manuscript is very powerful indeed. It is arguably that of the range of environmental concerns, climate change has taken too much of the attention. Many of the more local/regional effects will be just as important to the individuals concerned. This is especially true of air quality, or strong local pollutants that threaten food security – both of which are either modulated by the land surface or impact on it.**
This paper, for the first time, places emphasis on non-global pollutants, and it is revealing from much of the literature cited that the implications are likely to be large in many instances. Those who build the air quality and atmospheric tracer components of weather forecasts, regional climate models or even full Earth System Models will appreciate this manuscript, bringing the latest understanding of the terrestrial role into a single document.

This review is slightly different to usual because most papers have quantitative plots which can be assessed and studied in detailed, and then commented on. So really it is only possible to give an overview here. The authors can if they like, consider the points above and associated context-placing. As always with manuscripts, please read through carefully again – especially as now a break since submitting. The paper is very long, and so possibly look for any places where the writing can be tightened. As environmental science is evolving fast, it might be worth a quick, targeted literature search of any very recent 2018 papers on Scopus or the Web-of-Science. Otherwise, I think the document could be published almost in its current form.

*Reply: We will re-read the paper as suggested to avoid repetitions. As specified to reviewer R1, didactic sections will be moved to an appendix to shorten the manuscript. Moreover, we will also update the literature.

**Comment: A very small thing - the legends in Figure 2 are in small font – please make them slightly bigger.**

*Reply: Will be changed.
* * *

---

## Author Response (AR1)

Reviews and syntheses:
Influences of landscape structure and land uses on local to regional climate and air quality

**- Response to Referee reviews -**

We would like to thank both referees for their comments and suggestions. We tried to account for all of the suggestions as much as possible and submit a revised version of the manuscript.
You can find below a point-by-point reply to each of the referees comments.
Attached are two versions of the manuscript
- Marked with additions in blue and changes in red
- Non-marked final version, which might be easier to read.

**Referee # 1**
General comments
Summary: The authors aim to review how land use affects climate and air quality. They present a framework for categorizing land use, and apply it to different aspects of the earth system, in order to demonstrate relationships between biosphere, climate, and air quality. They also present current challenges to understanding the effects of land use on air quality, provide suggestions for different groups to work together to address these challenges.

I appreciate the effort to synthesize this information, and I generally agree with the abstract, but this paper falls very short of its goals. It is poorly organized, repetitive, inconsistent in its application of the framework, contains enough awkward language up front that the goals of the paper are not clear, and the conclusions do not follow from the information presented. Additionally, there isn't a unifying story to make sense of the extremely diverse information presented. For example, the air quality aspect is in the title and highlighted in the abstract, but is not addressed until page 23, and then only elaborated upon in a few sections. Also, main conclusions focus on model shortcomings, but there isn't a modelling review included.

It seems to me that this is about 5 papers mashed into one. Most of the text reads like a list, and links and meaning across the different sections and information are not made. The middle section alternates between pedantic textbook material and an uncritical presentation of many studies. I suggest that the authors think about what point they want to make, and focus on that point. For example, the material is here for a review of land use and air pollution. But there is a lot extra physics information that doesn't need to be presented in order make the point that the physical processes are an influence

*Reply:
We re-organized the manuscript to respond to the reviewers concerns according to the following:
- We reviewed the abstract so as to better reflect the objectives and the content of the manuscript
- We completed the introduction with supplementary literature related to land-use, land cover changes, and updated it. In the Introduction, we added that this review is intended to be accessible to the specialists (i.e., mainly scientists) and non-specialists (e.g., land-planners, stakeholders and decision-makers).
- We moved the "textbook material" highlighted in section 3 to an appendix to provide fundamentals of physics, chemistry and biology for those who may not be familiar with the theory that is behind the reviewed studies.
- We profoundly re-organized section 3. Rather than the "physical", chemical" and "biologeochemical" subsections further subdivided (repetitively) by land categories, we changed to two main sub-sections organized by land use and land cover changes: Land cover changes (deforestation, wetland conversion, urbanisation), Land intensification (agricultural intensification and urban intensification)
- Within each section, we separated the literature review according to physical, biological and chemical studies.
- We tried to include a critical summary the shortcomings. We used the conclusions (Section 5.1: synthesis of current knowledge) to back up the different sections.
We would like to express here that this is a substantial revision of our paper.

**Comment**
The introduction and framework are general and vague. Make a solid, but concise, assessment of land use/cover change as a foundation, with literature to back it up, and move on to the effects you want to review.
*Reply:
The assessment of land use/cover change is already in section 2 of the manuscript (rather than in the introduction). We added a section detailing the potential impacts of those changes on the atmospheric compartment (which is a very brief summary of sections 3.1.1, 3.2.1 and 3.3.1, the details of which are

moved to an appendix section). For us, the introduction is here to present the specificities of our review compared to other existing reviews in the literature and to state the objectives of this review.

**Comment**

The land categorization is applied inconsistently, probably due to its overlapping nature. Maybe delineate by urban, agriculture, and other lands. Part of the confusion and repetition arises because the urban changes are land cover changes, but these two have been separated.

*Reply:

The difficulty is in the fact that we are not looking at land use but at land use changes. We made the effort to limit the categories we look at regarding the Urban changes. We only have urban intensification now.

**Comment**

The physical, biological, chemical distinction does not work. The physical isn't related to the rest (except in 4.3 where is it relevant and sufficient), and the biological and chemical are both about biological emissions, with a fuzzy distinction between primary biogeochemistry (co2, ch4, n2o), and trace gases and aerosols. Stick to the chemical species you are interested in, and organize them around the land categories or land changes.

*Reply:

We agree with your analysis. We removed this distinction in our outline and re-organise the paper by categories of land-use changes, detailing for each category the impacts on the atmosphere and tried as much as possible to include a review of models. Thanks to this re-organisation, we are able to show the links between different studies and be more critical of the results based on other results.

However one objective of our paper is to clearly put forward the various ways land and atmosphere interact (via changes in physical processes, biological and chemical processes), and which are almost never considered together in climate models (nor global nor regional). This is clarified in the discussion.

**Comment**

Develop meaning and relationships through the presentation of the literature. Having separate discussions later, or pedantic explanations before, leaves the information essentially as a list, and the later discussions become repetitive and do not have the references to back up statements.

*Reply:

By re-organising the manuscript as explained in responses 1 and 3 and by moving the "textbook" material to an appendix, we think that the literature review is improved, less pedantic and more discursive, rather than a list. The section "synthesis of current knowledge" (5.1) is merged within section 3 to follow on directly and synthesize the review done. This way the discussion section will come earlier in the manuscript and is less repetitive.

**Comment**

To show integration of processes, put them in the same section. Present evidence for each one and the evidence for how they interact. Segregating them by section makes it difficult to make linkages without repetition. For example, section 4.3 starts to tie together vegetation, boundary layer, and air pollution, and is understandable without the lengthy textbook sections in part 3.

*Reply:

We agree with the reviewer and, as stated above (Response 4), we merged sections on physical, chemical and biological processes involved in land use changes and discuss their impacts in a more comprehensive way that looks at the interactive system as a whole.

**Comment**

To criticize models, you need a model review.

*Reply:

We added paragraphs in section 3 and within each land cover change section specific to modelling studies Our intention is not to criticize models but rather to show that there are today two 'niches' that have not been sufficiently looked at: 1) the combined physical-biological-chemical effects of land changes on climate at all spatial scales, 2) the specific 'territorial' scale that is smaller than the continental one and larger than a single city.

**Comment**
To make conclusions about what is lacking, the gaps and limitations of existing work need to be explained in the review, rather than listing all the literature results as facts. Currently, the paper reads like everything has been figured out, but the conclusions state that hardly anything has been figured out. The shortcomings of the models are not reviewed, but are presented as main conclusions.

*Reply:

We re-organised section 3 and tried having a more critical review of the literature, we present more clearly the gaps and limitations of models as well as experimental results, and move on directly to the discussion and conclusion sections.

Again our main argument is not to say that everything has been figured out nor that 'hardly anything has been figured out' but that the main pieces of the puzzle have not yet been assembled, nor applied at the specific spatial scale we're targeting.

**Specific comments/suggestions:**

**Comment**
Abstract (page 1)

physical, chemical, and biogeochemical land-atmosphere interactions is a very broad topic, while the paper focuses on air quality impacts of land use. the abstract needs to be clear about the focus of the paper.

*Reply:

The abstract was reformulated in order to make the focus of the paper clearer.
We are surprised though that the reviewer gets the impression we are focusing on air quality while discussion on this only occurs in $1/3^{rd}$ of the paper.

**Comment**
The focus on urban/peri-urban and air quality is not clear. until the last few sentences This leaves the reader wondering why the rapidly growing body of literature on the effects of LULCC on the earth system is not accounted for (only a few papers are cited in the intro).

*Reply:

We stated at the beginning of the introduction the specificities of the manuscript to justify the choice of literature cited.
As discussed above, there is no specific focus on air quality. Air quality is one of the 3 aspects we're targeting. Between our 2 reviewers, one has understood our point, not the other. We clarified this in the updated version.

**Comment**
page 2, line 7: Not sure that anthropized is a word. Anthropogenic seems correct, although not usually applied this way.

*Reply:

We do not believe "anthropogenic" is the correct term to be used. We replaced anthropized by man-shaped.

**Comment**
page 2, line 9: reference for energy balance?

*Reply:

A reference was added

**Comment**
section 2 and use and intensification - this does not seem to be the appropriate title for section 2 - section 2 covers a lot more than this

*Reply:

We changed the title to: Land Cover and Land Use changes: history, dynamics and challenges.
We also added text to this section explaining what is our understanding of agricultural intensification and urban intensification.

**Comment**
page 4 lines 21-23: the section should start with this. it is unclear why the land use section starts with land cover. see comment above.

*Reply:

We agree and changed accordingly.

**Comment**
section 2.2: not much definition here. in fact you acknowledge that definitions vary considerably
*Reply:
Title changed

**Comment**
page 5, lines 23-26: confusing- i am not sure what these numbers refer to.
*Reply:
These numbers are better explained.

**Comment**
page 6, lines 7-8: reference?
*Reply:
Reference is added.

**Comment**
not sure all these equations are necessary. this whole section seems like a textbook. who is the audience? a shorter description of how things change is more meaningful. The description can cite various studies on these effects, and be more digestible by the reader. there are no citations in this section. actually, this section can be deleted because the next 3 sections are the ones that makes the point.
*Reply:
This section was moved to an appendix. Moreover, in the Introduction.

**Comment**
this section makes the case of the previous section.
*Reply:
This section is moved to an appendix

**Comment**
page 12, line 29: and burning, and understory treatment, and different types of harvest, and planting
*Reply:
Sentence is added.

**Comment**
again, this section makes the point of section 3.1
*Reply:
This is addressed by changing the organisation.

**Comment**
section 3.1.3 urban intensification
this should probably be grouped with LULCC and it doesn't need all the references to the equations
*Reply:
References to equations are removed.
The section is modified as stated above in the general replies.

**Comment**
section 3.2 biological
this textbook section is unnecessary as the following sections make the case
*Reply:
Agree and is removed to an appendix.

**Comment**
3.3 chemical
another long textbook section
*Reply:
Agree and is removed to an appendix.

**Comment**
3.3.1 land use intensification (page 23)
here is a review relevant to land change and air quality. But it reads more like a list than a review of evidence for making a point.
*Reply:
Title is changed to land cover change. Content as stated above ie merged with other sections and is synthesized to read less as a list.

**Comment**
3.3.2 ag intensification
it seems like the previous ag section was cut short to put the information here
*Reply:
These sections are merged as stated above.

**Comment**
3.3.3 urban
page 23: what are the suggested trees in CA?
*Reply:
We specified which tree species are concerned.

**Comment**
4.1 local to meso and 4.2 ecosystem
these don't seem to have any relation to air quality
*Reply:
In this paper we focus on the effects of LULCC and LI on air quality but also on local climate. This is clarified in the introduction.

**Comment**
4.3 air quality
this is relevant, and expands upon sections 3.3.1-3.3.3 but now you have mashed together the land type framework
*Reply:
We agree with you, however in this section we have chosen to look at the interactions from a landscape/regional perspective and to discuss how differently LULUCC and LI interact together within a spatial framework.

**Comment**
5.1
this is a lengthy repetition of the previous reviews without the citations
*Reply:
This section is removed and merged within the new section 3 as stated above.

**Comment**
page 34
you really haven't shown the linkages and feedbacks. the information is all compart-mentalized. While your challenges are legitimate, it isn't clear how you reached them based on the previous reviews. there are little to no citations in this section, especially regarding modelling. you also don't acknowledge that models are essentially simplifications, and that they cannot contain every single detail, and that not every single detail matters for the desired outcome of a model.
*Reply:
By changing and better discussing the literature in the modified section 3 we believe this question is addressed. We acknowledged model specificities as recommended.

5.3 interdisciplinary approaches
this is repetitive
*Reply:
We understand that the reviewer refers to the introduction of section 5.3. We reviewed this part and removed repetitive ideas.

**Comment**

6 bridging the gap

while this is important, i don't don't see how it relates to this article

*Reply:

In this section, we treat existing gaps between scientific communities and spatial planners, which is justified, in our view since we need models that are more robust and a more exhaustive representation of scenarios that account for the social and economic drivers in the system.

On the other hand, we agree that the part focusing on policy makers is not in the focus of this review and we removed it.

**Comment**

7 conclusion

there isn't a modelling review, so i don't know how presenting all of the field research generates the conclusion that modelling is poor

*Reply:

We rephrased the conclusion.

**Referee #2**

**Comment**

Thank you for inviting me to review this paper. First and foremost, I would like to state that it is one of the most comprehensive manuscripts I have seen for some time. It has the potential to be a significant "go to" paper for anyone with interest in measuring or modelling land-atmosphere interactions. The reference list is – as might be expected for a review – very thorough, and it certainly alerted me to papers I was not aware of.

I think it is important first to note what the paper does not include. This is not to detract from it in any way at all, but simply to gain understanding as to what its main messages are. The authors may like to consider a set of words to capture this, possibly towards the end of the Introduction. So not addressed in detail are:

(1) Many existing studies focus on the role of the land surface in mitigating carbon dioxide emissions. Significant effort is placed in closing the global carbon cycle, and there is a view that the land surface (and associated land-atmosphere $CO_2$ exchanges under a changing climate) is where much uncertainty remains. In the most general terms, approximately 25% of $CO_2$ emissions are believed to be drawn-down by the terrestrial ecosystems. Of concern is that this fraction may decrease into the future, especially through higher respirations or nutrient limitation. There is a small reference to this, indirectly, in Table 2 "Change in atmospheric concentrations of GHG". The title is clear, with no word "global" used, but it does mention "climate".

*Reply:

In this review we choose to treat the effects on air quality and local climate. The impact on the global climate (GHG emissions) is not the focus of our study. We added an explicit phrase in the introduction to better explain this point.

**Comment**

(2) Related to (1) above, much is described in the IPCC reports, and especially the recent 1.5°C and 2.0°C threshold assessment, about the role of BioEnergy with Carbon, Capture and Storage (BECCS). This form of large-scale geo-engineering of the global carbon cycle is not included here (for instance, the $CO_2$ cycle is omitted from Schematic Figure 5).

*Reply:

Again as for global climate, the C cycle is not the focus of our study. This topic has been already extensively reviewed, even though we agree that there are still a lot of gaps and challenges ahead. We specifed this point in the Introduction and refer the interested readers to recent reviews (e.g., Le Quere et al., 2018; Saunois et al., 2018).

**Comment**

(3) The paper is very much a qualitative assessment, with most display items more schematic in format. Hence, it is presently difficult to compare effects, and so the logical conclusion is the one that the authors present. That is, there is a need for an overall integrated tool that would allow effective intercomparison of regional effects, drivers and feedbacks.

*Reply:

We agree with this analysis, and, upon suggestion from reviewer R1, we have re-organised the manuscript so as to better show how those conclusions are reached.

**Comment**
(4) The major part of the paper concerns geochemical feedbacks, rather than the more physical one. There are some exceptions. For instance, one page 14, there are citations to papers describing how different land cover types have the potential to either suppress or aggravate any future extremes in a changed climate.

*Reply:
These citations are presented in the section talking about physical effects. This section is merged with others according to Land Cover Changes (LCC) and Land Intensification (LI). In this sense, we try to homogenize with other effects and consider if this is a point that should be more thoroughly reviewed or removed from our manuscript

**Comment**
By stating something along the lines (1) – (4) will then make the paper stronger, as clearer then what the paper does encompass. Moreover, this is where I believe the manuscript is very powerful indeed. It is arguably that of the range of environmental concerns, climate change has taken too much of the attention. Many of the more local/regional effects will be just as important to the individuals concerned. This is especially true of air quality, or strong local pollutants that threaten food security – both of which are either modulated by the land surface or impact on it.

This paper, for the first time, places emphasis on non-global pollutants, and it is revealing from much of the literature cited that the implications are likely to be large in many instances. Those who build the air quality and atmospheric tracer components of weather forecasts, regional climate models or even full Earth System Models will appreciate this manuscript, bringing the latest understanding of the terrestrial role into a single document.

This review is slightly different to usual because most papers have quantitative plots which can be assessed and studied in detailed, and then commented on. So really it is only possible to give an overview here. The authors can if they like, consider the points above and associated context-placing. As always with manuscripts, please read through carefully again – especially as now a break since submitting. The paper is very long, and so possibly look for any places where the writing can be tightened. As environmental science is evolving fast, it might be worth a quick, targeted literature search of any very recent 2018 papers on Scopus or the Web-of-Science. Otherwise, I think the document could be published almost in its current form.

*Reply:
We re-read the paper as suggested and improved parts that were considered as repetitive. As specified to reviewer R1, didactic sections are moved to an appendix to shorten the manuscript. Moreover, we also updated the literature.

**Comment**
A very small thing - the legends in Figure 2 are in small font – please make them slightly bigger.
*Reply:
This is changed.

**Reviews and syntheses: Influences of landscape structure and land uses on local to regional climate and air quality**

Raia Silvia Massad[1], Juliette Lathière[2], Susanna Strada[3], Mathieu Perrin[4], Erwan Personne[1], Marc Stefanon[5], Patrick Stella[4], Sophie Szopa[2], Nathalie de Noblet-Ducoudré [2]

1 UMR ECOSYS, INRA AgroParisTech, Université Paris Saclay, 78850, Thiverval Grignon, France
2 Laboratoire des Sciences du Climat et de l'Environnement, LSCE/IPSL, CEA-CNRS-UVSQ, Université Paris-Saclay, Gif-sur-Yvette, 91191, France
3 LMD/IPSL, Ecole polytechnique, Université Paris-Saclay, Sorbonne, Universités UPMC Univ. Palaiseau France
 The Abdus Salam International Centre for Theoretical Physics - Earth System Physics Section, 34151 Trieste, Italy
4 UMR SAD-APT, AgroParisTech, INRA, Université Paris-Saclay, 75005, Paris, France
5 LMD/IPSL, Ecole polytechnique, Université Paris-Saclay, Sorbonne, Universités UPMC Univ. Palaiseau France

*Correspondence to*: Raia Silvia Massad (raia-silvia.massad@inra.fr)

**Abstract**. The atmosphere and the land surface interact in multiple ways, for instance through the radiative-energy balance, the water cycle or the emission-deposition of natural and anthropogenic compounds. By modifying the land surface, land-use and land-cover changes (LULCCs) and land management changes (LMCs) alter the physical, chemical and biological processes of the biosphere and therefore all land-atmosphere interactions, from local to global scales. Through socio-economic drivers and regulatory policies adopted at different levels (local, regional, national or supranational), human activities strongly interfere in the land-atmosphere interactions, and those activities lead to a patchwork of natural, semi-natural, agricultural, urban and semi-urban areas. In this context, urban and peri-urban areas, which have a high population density, are of particular attention since land transformation can lead to important environmental impacts and affect the health and life of millions of people. The objectives of this review is to synthesize the existing experimental and modelling works that investigate physical, chemical and/or biogeochemical interactions between land surfaces and the atmosphere, therefore potentially impacting local/region climate and air quality, mainly in urban or peri-urban landscapes at regional and local scales.

The conclusions we draw from our synthesis are the following. (1) The adequate temporal and spatial description of land-use and land-management practices (e.g. areas concerned, type of crops, whether or not they are irrigated, quantity of fertilizers used and actual seasonality of application) necessary for including the effects of LMC in global and even more in regional climate models is inexistent (or very poor). Not taking into account these characteristics may bias the regional projections used for impact studies. (2) Land-atmosphere interactions are often specific to the case study analysed; therefore, one

Commenté [Coauthors1]: Based on Susanna Strada's strong contribution to the revision of the manuscript, coauthors agreed to upgrade her as a 3rd author. The list and affiliation order of the coauthors have been updated accordingly.

[revised manuscript text omitted]

**1.**

**Commenté [Coauthors3]:** This section has been intensively reorganized, based on the reviewers feedbacks. The theoretical sections (physical/biological/chemical) have been moved to the appendices and the presentation of existing studies have been structured by land cover and land intensification categories rather than by processes, in order to give a more comprehensive and complementary view of the various impacts.

[revised manuscript text omitted]